# Identification of IMC43, a novel IMC protein that collaborates with IMC32 to form an essential daughter bud assembly complex in *Toxoplasma gondii*

**Rebecca R. Pasquarelli[1], Peter S. Back[1], Jihui Sha[2], James A. Wohlschlegel[2], Peter J. Bradley** [1,3] *

**1** Molecular Biology Institute, University of California, Los Angeles, California, United States of America,
**2** Department of Biological Chemistry and Institute of Genomics and Proteomics, University of California, Los Angeles, California, United States of America, **3** Department of Microbiology, Immunology, and Molecular Genetics, University of California, Los Angeles, California, United States of America

* pbradley@ucla.edu

**Data Availability Statement:** All relevant data are within the manuscript and its Supporting Information files.

## Abstract

The inner membrane complex (IMC) of *Toxoplasma gondii* is essential for all phases of the parasite's life cycle. One of its most critical roles is to act as a scaffold for the assembly of daughter buds during replication by endodyogeny. While many daughter IMC proteins have been identified, most are recruited after bud initiation and are not essential for parasite fitness. Here, we report the identification of IMC43, a novel daughter IMC protein that is recruited at the earliest stages of daughter bud initiation. Using an auxin-inducible degron system we show that depletion of IMC43 results in aberrant morphology, dysregulation of endodyogeny, and an extreme defect in replication. Deletion analyses reveal a region of IMC43 that plays a role in localization and a C-terminal domain that is essential for the protein's function. TurboID proximity labelling and a yeast two-hybrid screen using IMC43 as bait identify 30 candidate IMC43 binding partners. We investigate two of these: the essential daughter protein IMC32 and a novel daughter IMC protein we named IMC44. We show that IMC43 is responsible for regulating the localization of both IMC32 and IMC44 at specific stages of endodyogeny and that this regulation is dependent on the essential C-terminal domain of IMC43. Using pairwise yeast two-hybrid assays, we determine that this region is also sufficient for binding to both IMC32 and IMC44. As IMC43 and IMC32 are both essential proteins, this work reveals the existence of a bud assembly complex that forms the foundation of the daughter IMC during endodyogeny.

## Author summary

*Toxoplasma gondii* is an obligate intracellular parasite that causes disease in immunocompromised individuals and congenitally infected neonates. *Toxoplasma* replicates using a unique process of internal budding in which two daughter buds form inside a single

**Funding:** This work was supported by NIH grants AI123360 to P.J.B. and GM089778 to J.A.W. R.R.P and P.S.B were supported by the Ruth L. Kirschstein National Research Service Award GM007185 and UCLA Molecular Biology Institute (MBI) Whitcome Fellowship. R.R.P. was additionally supported by the Ruth L. Kirschstein National Research Service Award AI007323. The funders had no role in study design, data collection and analysis, decision to publish, or preparation of the manuscript.

**Competing interests:** The authors have declared that no competing interests exist.

maternal parasite. This process is facilitated by an organelle called the inner membrane complex (IMC) which is found in *Toxoplasma* and other parasites in the phylum Apicomplexa. Although the IMC is known to be required for parasite replication, only a few early-recruiting IMC proteins have been identified. In this study, we identify and functionally analyze a novel IMC protein which is one of the earliest components of daughter buds and plays an essential role in parasite replication. We also identify its binding partners and demonstrate how their interactions impact the construction of daughter cells. This work reveals the existence of an essential protein complex formed in the earliest stages of parasite replication, expands our understanding of the IMC's role in *Toxoplasma* replication, and identifies potential targets for therapeutic intervention.

## Introduction

The Apicomplexa are a phylum of obligate intracellular parasites which cause serious disease in both humans and animals worldwide, leading to significant morbidity and mortality as well as economic losses [1]. The phylum includes the human pathogens *Toxoplasma gondii* (toxoplasmosis), *Plasmodium spp.* (malaria), and *Cryptosporidium spp.* (diarrheal diseases) as well as the veterinary pathogens *Eimeria tenella* (chicken coccidiosis) and *Neospora caninum* (neosporosis) [2–6]. *T. gondii* infects roughly 30% of the global human population [7]. While infection is typically asymptomatic in healthy individuals, infection in immunocompromised individuals can lead to fatal encephalitis [8,9]. In addition, congenital toxoplasmosis derived from a primary maternal infection can cause severe fetal abnormalities and abortion [10]. Existing treatments are restricted to limiting acute disease and are poorly tolerated [9]. Thus, a deeper understanding of the biology of this important pathogen is critically needed for the development of therapies based on parasite-specific activities.

The life cycles of apicomplexan parasites depend on a specialized organelle named the inner membrane complex (IMC), which plays critical roles in motility, host cell invasion, and parasite replication [11]. In *T. gondii*, the IMC lies directly underneath the parasite's plasma membrane and is composed of two structures: a series of flattened vesicles called alveoli which are arranged in a quilt-like pattern and a supportive network of intermediate filament-like proteins called alveolins [12,13]. The IMC is further divided into distinct regions such as the cone-shaped apical cap at the apex of the parasite, a central body portion, and a ring-shaped basal complex at the basal end of the parasite. Detergent fractionation experiments have shown that IMC proteins can associate with the alveoli, the cytoskeletal network, or span between both layers [14–16]. Additionally, IMC proteins differ by their presence in the maternal IMC, daughter IMC, or both [14,16]. Underlying the IMC is an array of 22 subpellicular microtubules that provide additional structural support for the organelle.

One of the key functions of the IMC is to facilitate parasite replication. Apicomplexans replicate using a variety of different budding mechanisms, all of which rely on the IMC to act as a scaffold for the developing daughter buds [17,18]. *T. gondii* replicates asexually by endodyogeny, a process of internal budding in which two daughter buds develop within the cytoplasm of a single maternal parasite. Budding is initiated upon centrosome duplication, which is quickly followed by the sequential recruitment of IMC proteins to the daughter cell scaffold (DCS) [14,18–21]. As proteins are added to the DCS, polymerization of subpellicular microtubules underlying the daughter IMC drives elongation of the growing daughter buds [22,23]. In the final stages of bud maturation, the maternal IMC is degraded and the daughter cells adopt the maternal plasma membrane and emerge as two separate cells [11].

The synthesis and recruitment of IMC proteins to the DCS occurs in a "just in time" manner [24,25]. As such, IMC proteins can be categorized by their timing of recruitment. During bud initiation the daughter IMC proteins IMC29, IMC32, FBXO1, BCC0, and IMC15 are recruited to the DCS [14,26–29]. IMC32 and BCC0 have been shown to be essential for parasite replication and survival, as their loss leads to an inability to stably assemble the IMC [27,28]. IMC29 and FBXO1, while not essential, also play critical roles in supporting the organization and maturation of the daughter IMC during endodyogeny [26,29]. The essential apical cap proteins AC9 and AC10 also localize to the daughter IMC during bud initiation, but these proteins are maintained in the maternal IMC and are involved in host cell invasion, rather than replication [30,31]. After bud initiation is completed, additional proteins such as ISP1 and the alveolins are added sequentially to the growing buds [14,32]. Though many protein components of the DCS have recently been identified, the organization, regulation, and precise function of most of these proteins remain poorly understood.

In this study, we identified and functionally analyzed a novel IMC protein, IMC43. We determined that IMC43 localizes to the body of the daughter IMC and is recruited during bud initiation. Using an auxin-inducible degron (AID) approach, we conditionally depleted IMC43 and showed that loss of the protein results in severe morphological defects, dysregulation of endodyogeny, and loss of overall lytic ability. We then performed deletion analyses which established that the function of IMC43 is dependent on a small region towards the C-terminus of the protein. Next, we used IMC43 as bait in both a TurboID proximity labelling experiment and a yeast two-hybrid (Y2H) screen to identify candidate IMC43 binding partners and provide insight into the precise function of the protein. Two of these proteins, the essential daughter protein IMC32 and a hypothetical protein we named IMC44, were functionally analyzed in relation to IMC43. These experiments showed that IMC43 is required for the proper localization of both of its partners and that binding at the essential C-terminal region of IMC43 is responsible for these interactions. Overall, our work identifies IMC43 as a foundational component of the early daughter IMC and reveals the existence of an IMC43-IMC32 daughter bud assembly complex which is essential for parasite replication and survival.

## Results

### IMC43 is a component of the early daughter cell scaffold

We have previously conducted proximity labelling to identify novel IMC proteins. In one of these experiments in which we used IMC29 as bait, we identified the uncharacterized protein TGGT1_238895 [29]. TGGT1_238895 encodes a 1,653 amino acid hypothetical protein which contains no identifiable functional domains (Figs 1A and S1A) [33,34]. We selected TGGT1_238895 for further analysis because of its extremely low phenotype score of -5.41 in a genome-wide CRISPR/Cas9 screen for fitness-conferring genes and its cyclical expression profile, which is similar to IMC29 (Fig 1B) [24,35].

To determine the localization of TGGT1_238895, we endogenously tagged it with a C-terminal 3xHA epitope tag. The protein was undetectable by immunofluorescence assay (IFA) in mature parasites, but colocalized with the IMC body-localizing alveolin IMC6 in developing daughter buds (Fig 1C and 1D). To determine whether TGGT1_238895 localized to the daughter apical cap, we co-stained with the apical cap marker ISP1. TGGT1_238895 was found to be absent from the apical cap, indicating that it is exclusively an IMC body protein (Fig 1E). Based on its localization, we named the protein IMC43. To determine how early IMC43 is incorporated into developing daughter buds, we co-stained for other early daughter markers that have been previously studied. These experiments showed that IMC43 recruits to early

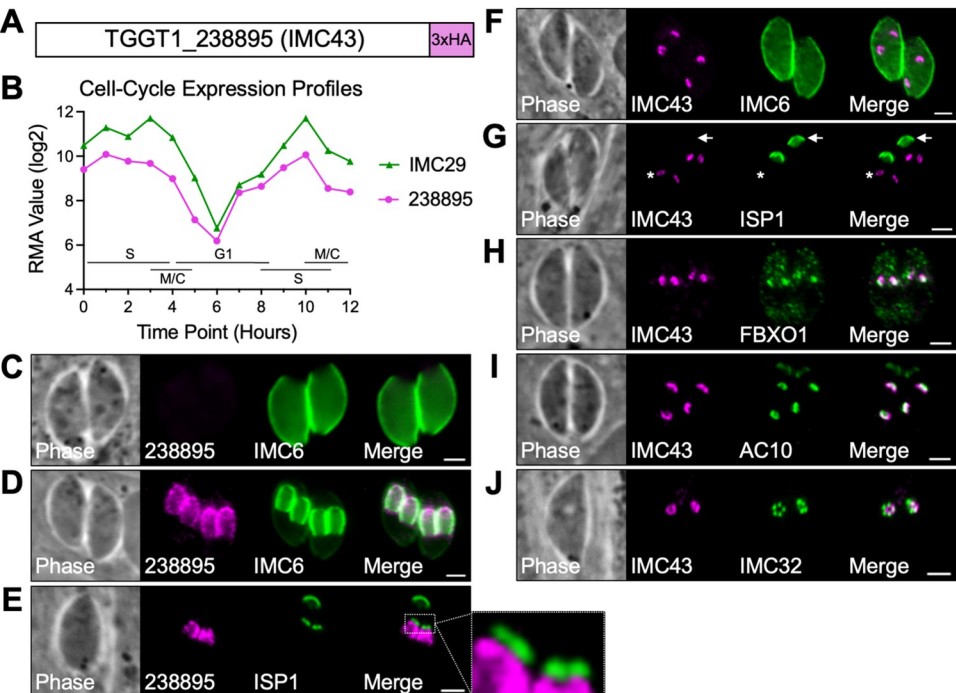

**Fig 1. TGGT1_238895 localizes to the daughter IMC body and is recruited at the earliest stages of endodyogeny.**
A) Gene model of TGGT1_238895 (IMC43) showing the endogenous 3xHA tag fused to the C-terminus for localization studies. B) The cell-cycle expression profile for TGGT1_238895 closely matches the cyclical pattern of the known daughter IMC protein IMC29. RMA = robust multi-array average [24]. C) IFA showing that TGGT1_238895 is undetectable in mature parasites. Magenta = anti-HA detecting TGGT1_238895[3xHA], Green = anti-IMC6. D) IFA showing that TGGT1_238895 localizes to daughter buds and colocalizes with IMC6. Magenta = anti-HA detecting TGGT1_238895[3xHA], Green = anti-IMC6. E) IFA of budding parasites showing that TGGT1_238895 is excluded from the apical cap, marked by ISP1. Magenta = anti-HA detecting TGGT1_238895[3xHA], Green = anti-ISP1. F) IFA showing that IMC43 recruits to the daughter IMC before IMC6. Magenta = anti-HA detecting IMC43[3xHA], Green = anti-IMC6. G) IFA showing that IMC43 recruits to the daughter IMC before ISP1. Arrow points to the mother cell apical cap. Asterisk indicates daughter buds. Magenta = anti-HA detecting IMC43[3xHA], Green = anti-ISP1. H) IFA showing that IMC43 recruits to the daughter IMC at the same time as FBXO1. Magenta = anti-Ty detecting IMC43[2xStrep3xTy], Green = anti-HA detecting FBXO1[3xHA]. I) IFA showing that IMC43 recruits to the daughter IMC at the same time as AC10. Magenta = anti-HA detecting IMC43[3xHA], Green = anti-Ty detecting AC10[2xStrep3xTy]. J) IFA showing that IMC43 recruits to the daughter IMC at the same time as IMC32. Magenta = anti-HA detecting IMC43[3xHA], Green = anti-V5 detecting IMC32[3xV5]. Scale bars = 2 μm.

daughters before IMC6 and ISP1 and around the same time as FBXO1, AC10, and IMC32 (Fig 1F–1J). From these data we concluded that IMC43 is a daughter-specific IMC body protein that is recruited to the DCS at the earliest stages of endodyogeny.

## IMC43 is an essential protein involved in endodyogeny

Because IMC43 has a low phenotype score and is a component of the early daughter cell scaffold, we hypothesized that it was likely to play a critical role in parasite survival and replication [35]. We were unable to genetically disrupt *IMC43*, despite successful CRISPR/Cas9 targeting to its endogenous locus, indicating that *IMC43* is likely essential as suggested by its phenotype score (S2 Fig). Thus, we used an auxin-inducible degron (AID) conditional knockdown system to study its function (Fig 2A) [36,37]. The degron-tagged protein (IMC43[AID]) localized correctly to the daughter IMC and was rapidly degraded upon treatment with indoleacetic acid (IAA) (Fig 2B). Western blot analysis confirmed efficient knockdown of the target protein (S3A Fig).

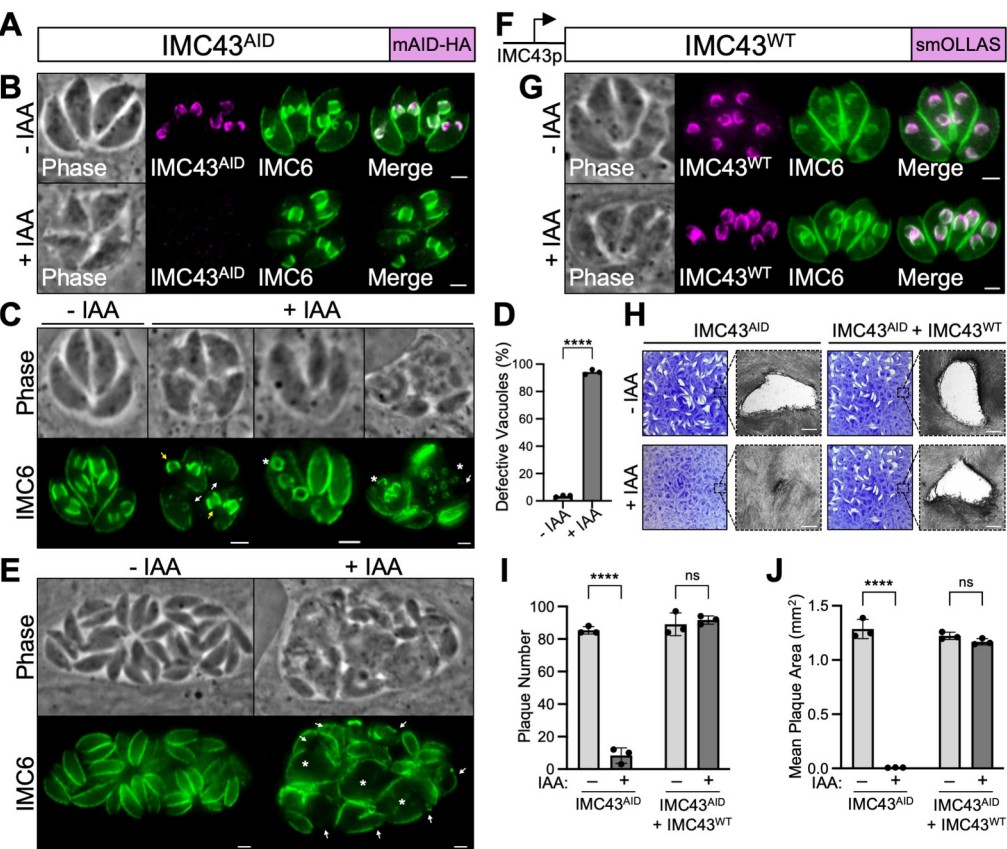

**Fig 2. IMC43 is essential for parasite replication and survival.** A) Diagram showing the mAID-3xHA degron tag fused to the C-terminus of IMC43 in a TIR1-expressing strain to facilitate proteasomal degradation upon treatment with IAA. B) IFA showing that the IMC43[AID] protein localizes normally to the daughter IMC and is depleted when the parasites are treated with IAA. Magenta = anti-HA detecting IMC43[AID], Green = anti-IMC6. C) IFA showing the broad range of morphological and replication defects observed after treating IMC43[AID] parasites with IAA for 24 hours. White arrows point to large gaps in the IMC marked by IMC6. Yellow arrows point to daughter buds present outside of the maternal parasite. Asterisks mark parasites producing more than two daughter buds. All three +IAA vacuoles also display desynchronized endodyogeny, where parasites in the same vacuole are at different stages of replication. Green = anti-IMC6. D) Quantification of vacuoles displaying morphological and/or replication defects after 24 hours of IMC43 depletion. Statistical significance was determined using a two-tailed t test (****, P < 0.0001). E) IFA of IMC43[AID] parasites treated with IAA for 40 hours. White arrows point to large gaps in the IMC marked by IMC6. Asterisks indicate swollen parasites. Green = anti-IMC6 F) Diagram of the full-length smOLLAS-tagged IMC43 complementation construct driven by its endogenous promoter (IMC43[WT]) and integrated at the UPRT locus in IMC43[AID] parasites. G) IFA showing that IMC43[WT] localizes normally to the daughter IMC and rescues the morphological and replication defects observed upon treatment with IAA. Magenta = anti-OLLAS detecting IMC43[WT], Green = anti-IMC6. H) Plaque assays for IMC43[AID] and IMC43[AID] + IMC43[WT] parasites grown for seven days -/+ IAA. Depletion of IMC43 results in a severe defect in overall lytic ability, which is fully rescued by complementation with the wild-type protein. Scale bars = 0.5 mm. I) Quantification of plaque number for plaque assays shown in panel H. IMC43-depleted parasites form fewer than 10% as many plaques compared to control. Statistical significance was determined using multiple two-tailed t tests (****, P < 0.0001; ns = not significant). J) Quantification of plaque size for plaque assays shown in panel H. Plaques formed by IMC43-depleted parasites are <1% the usual size. Statistical significance was determined using multiple two-tailed t tests (****, P < 0.0001; ns = not significant). Scale bars for all IFAs = 2 μm.

To assess the effects of IMC43 knockdown on parasite morphology over the course of multiple replication cycles, parasites were treated with IAA for 24 hours and assessed by IFA (Fig 2C). IMC43-depleted parasites showed severe defects in IMC morphology, such as large gaps in the cytoskeleton marked by IMC6 and the presence of daughter buds that appeared to not be encased by the maternal IMC. IMC43-depleted parasites also exhibited a variety of

replication defects, including asynchronous division and formation of more than two daughter buds per cell, indicating a severe dysregulation of endodyogeny. Quantification showed that 94.2% of vacuoles exhibited one or more of these issues after only 24 hours of IAA treatment (Fig 2D), with most vacuoles exhibiting multiple defects simultaneously. When we extended the IAA treatment to 40 hours, we saw continued growth despite these defects, which appeared to accumulate over time (Fig 2E). To confirm that these defects were due to the loss of IMC43, we generated a full-length IMC43 complementation construct driven by its endogenous promoter and integrated it at the UPRT locus (IMC43$^{WT}$) (Figs 2F and S3B). Complementation with IMC43$^{WT}$ fully rescued the observed morphological and replication defects (Fig 2G).

To determine how loss of IMC43 affects the parasite's overall lytic ability on a longer time course, we performed plaque assays (Fig 2H). IMC43-depleted parasites exhibited a severe defect in plaque efficiency, forming <10% as many plaques compared to vehicle-treated IMC43$^{AID}$ parasites (Fig 2I). The few plaques that they did form were also drastically smaller than normal: <1% the size of plaques formed by vehicle-treated parasites (Fig 2J). The defects in both plaque efficiency and plaque size were fully rescued by complementation with the wild-type protein. Since we were unable to generate a stable knockout line, it's likely that these few small plaques are formed by a small percentage of parasites that experience incomplete knockdown.

To dissect the phenotype of IMC43-depleted parasites further, we co-stained for markers of key structures involved in endodyogeny. First, we stained for the daughter IMC body protein IMC29, which localized normally despite the gross morphological defects (Fig 3A). Next, we stained for the apical cap marker AC10, the basal complex marker MORN1, and tubulin. These experiments showed that the apical cap, basal complex, conoid, and subpellicular microtubules appear to assemble on nascent daughter buds in the absence of IMC43 (Fig 3B–3D). Finally, we stained for the centrosome marker Centrin1, as well as DNA marked by Hoechst staining. This showed that in the absence of IMC43, parasites continue nuclear division and centrosome duplication in an apparent attempt to continue replicating (Fig 3E and 3F). Together, these data indicate that IMC43 is an essential protein which is required for replication, structural integrity of the IMC, and parasite survival.

## IMC43 contains an essential functional domain near the C-terminus of the protein

Like many other IMC proteins, IMC43 lacks homology to known proteins and has no readily identifiable functional domains. To dissect which regions of the protein are important for localization and function, we developed a series of nine deletion constructs guided by predicted secondary structure and conservation with *N. caninum* (Figs 4A and S1B) [38]. Each deletion construct was integrated into the UPRT locus in the IMC43$^{AID}$ strain, as we did previously for IMC43$^{WT}$. Western blot analysis confirmed that each of the deletion constructs was expressed at similar levels to IMC43$^{WT}$ and ran at approximately the expected size (S3C Fig). Eight of the nine deletion constructs colocalized completely with the wild-type protein (S4 Fig). The only deletion construct which did not was IMC43$^{\Delta 402-494}$, which enriched at daughter buds but also partially mistargeted to the cytoplasm (Fig 4B). Surprisingly, almost all the deletion constructs were also able to rescue both the morphological and replication defects. The only exception was the Δ1441–1653 mutant, a C-terminal truncation of the protein (Fig 4C and 4D, S5). To narrow down the essential region of the C-terminus, we constructed two smaller sub-deletions: Δ1441–1561 and Δ1562–1653. The Δ1562–1653 mutant fully rescued the phenotype of the knockdown, while the Δ1441–1561 mutant did not, indicating that residues 1441–1561 are essential for the function of IMC43 (Figs 4E–4G and S3D).

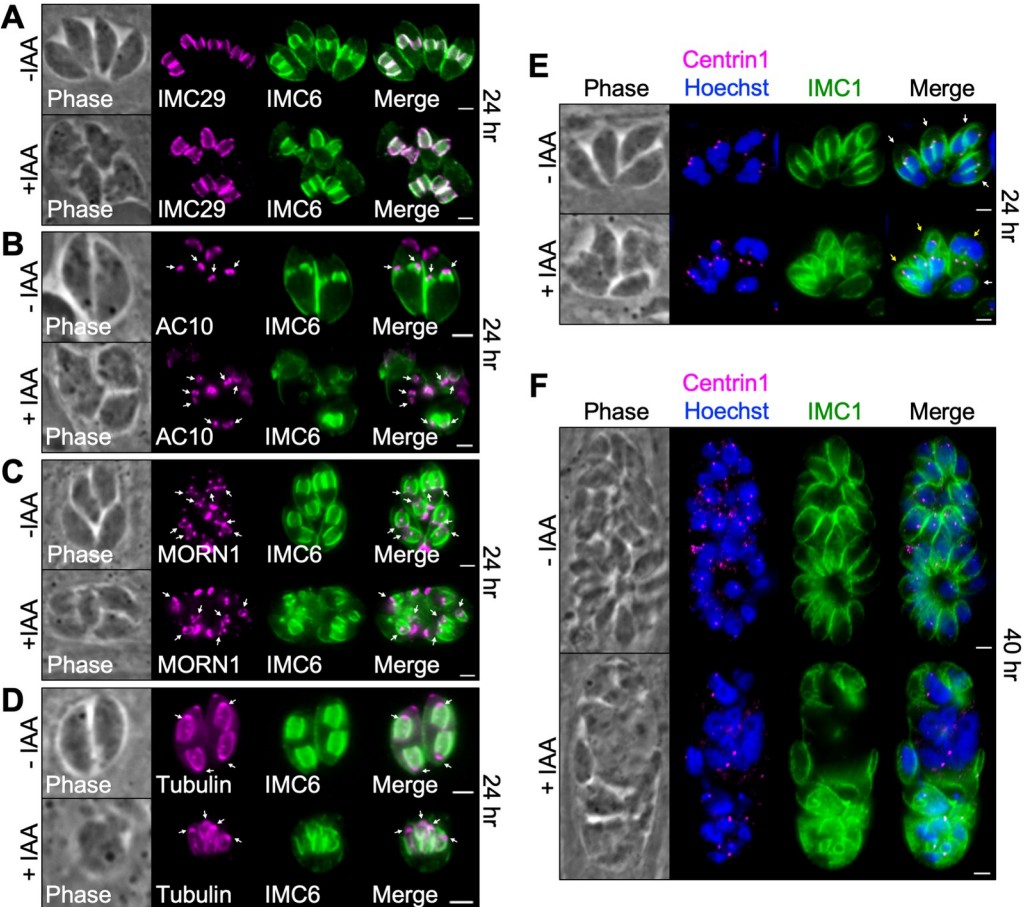

**Fig 3. Assessment of key structures involved in endodyogeny.** A) IFA showing that the daughter IMC protein IMC29 is unaffected by depletion of IMC43. Magenta = anti-V5 detecting IMC29$^{3xV5}$, Green = anti-IMC6. B) IFA showing that the apical cap marker AC10 is unaffected by depletion of IMC43. Arrows point to the apical cap of daughter buds. Magenta = anti-V5 detecting AC10$^{3xV5}$, Green = anti-IMC6. C) IFA showing that the basal complex marker MORN1 is unaffected by depletion of IMC43. Arrows point to the basal complex of daughter buds. Magenta = anti-V5 detecting MORN1$^{3xV5}$, Green = anti-IMC6. D) IFA showing that the conoid and subpellicular microtubules still assemble on nascent daughter buds when IMC43 is depleted. Arrows point to daughter bud conoids. Magenta = transiently expressed Tubulin1-GFP, Green anti-IMC6. E) IFA showing that centrosome duplication continues when IMC43 is depleted. Centrosomes, marked by Centrin1, appear to duplicate and associate with daughter buds and dividing nuclei. Parasites forming more than two daughter buds also form more than two centrosomes. White arrow points to a single parasite containing two centrosomes (normal). Yellow arrows point to single parasites containing three or more centrosomes (abnormal). Magenta = anti-Centrin1, Green = anti-IMC1, Blue = Hoechst. F) IFA showing that nuclear division and centrosome duplication continue after 40 hours of IAA treatment. Magenta = anti-Centrin1, Green = anti-IMC1, Blue = Hoechst. Scale bars = 2 μm.

## Identification of IMC43 binding partners using TurboID and Y2H screening

To further investigate IMC43, we used both TurboID proximity labelling and a yeast two-hybrid (Y2H) screen to identify candidate binding partners [39–41]. For the TurboID experiment, we fused the TurboID biotin ligase plus a 3xHA tag to the C-terminus of IMC43 (Fig 5A). IMC43$^{TurboID}$ localized normally and robustly biotinylated the daughter IMC when treated with biotin, confirming that the TurboID biotin ligase is enzymatically active (Fig 5B). Therefore, we continued with a large-scale TurboID experiment and analyzed the results by mass spectrometry. The results were filtered to include only genes that were at least two-fold

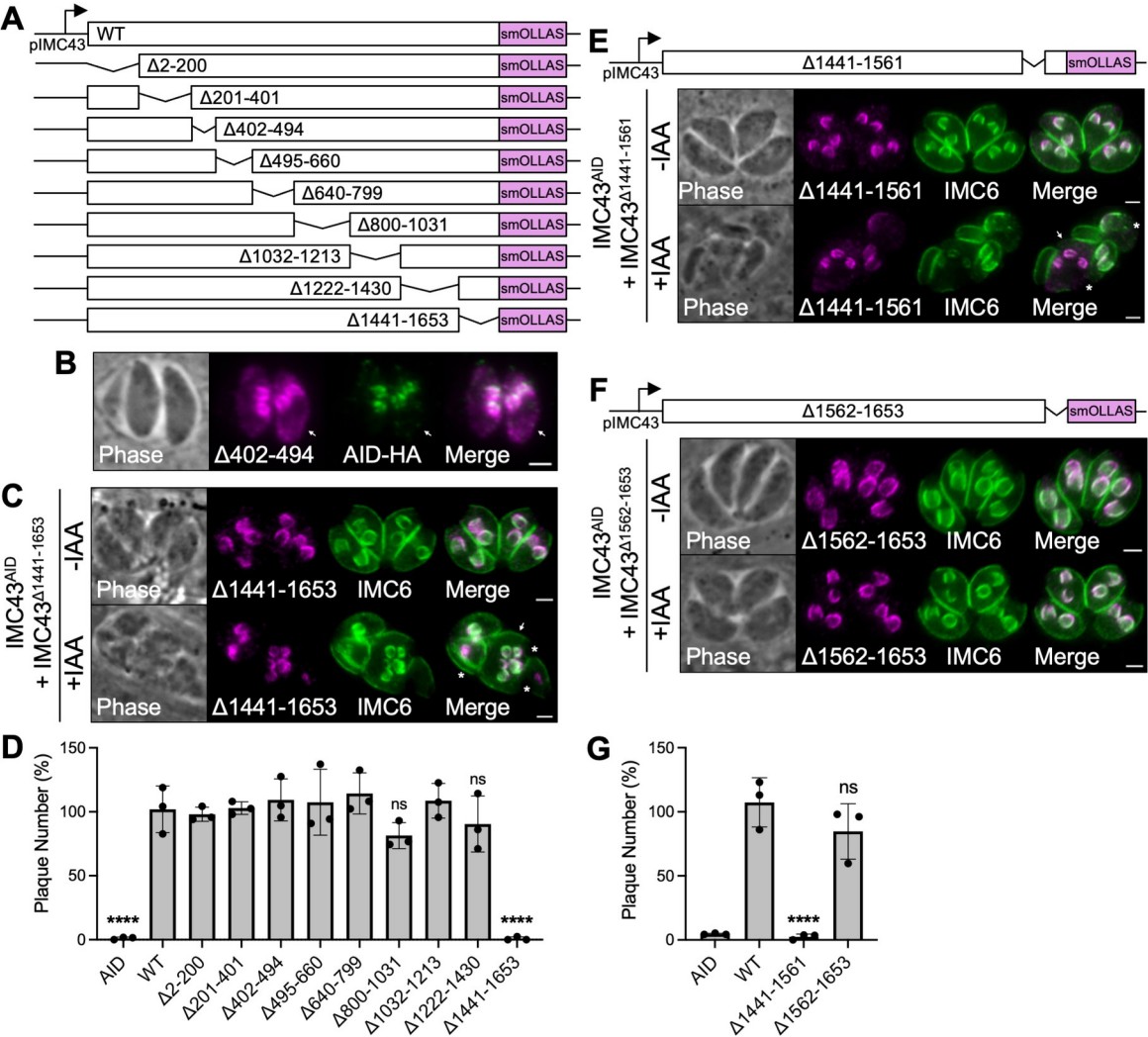

**Fig 4. Deletion analyses reveal regions involved in IMC43 localization and function.** A) Diagram of the nine deletion constructs generated for domain analysis of IMC43. B) IFA showing the localization of IMC43$^{\Delta402-494}$ compared to IMC43$^{AID}$. IMC43$^{\Delta402-494}$ partially mislocalizes to the cytoplasm of parasites (arrow). Magenta = anti-OLLAS detecting IMC43$^{\Delta402-494}$, Green = anti-HA detecting IMC43$^{AID}$. C) IFA showing that IMC43$^{\Delta1441-1653}$ fails to rescue the morphological and replication defects caused by depletion of endogenous IMC43. Arrow points to a single maternal parasite producing five daughter buds, indicating dysregulation of endodyogeny. Asterisks indicate disruption of the cytoskeleton. Magenta = anti-OLLAS detecting IMC43$^{\Delta1441-1653}$, Green = anti-IMC6. D) Quantification of plaque number for IMC43$^{AID}$, IMC43$^{AID}$ + IMC43$^{WT}$, and all nine IMC43 deletion strains shown in panel A after seven days of growth -/+ IAA. Y axis represents the number of plaques formed in +IAA conditions divided by the number of plaques formed in -IAA conditions for each strain. Statistical significance was determined by one-way ANOVA (****, P < 0.0001; ns = not significant). E) IFA showing that IMC43$^{\Delta1441-1561}$ fails to rescue the morphological and replication defects caused by depletion of endogenous IMC43. Arrow points to a single maternal parasite producing three daughter buds, indicating dysregulation of endodyogeny. Asterisks indicate disruption of the cytoskeleton. Magenta = anti-OLLAS detecting IMC43$^{\Delta1441-1561}$, Green = anti-IMC6. F) IFA showing that IMC43$^{\Delta1562-1653}$ rescues the morphological and replication defects caused by depletion of endogenous IMC43. Magenta = anti-OLLAS detecting IMC43$^{\Delta1562-1653}$, Green = anti-IMC6. G) Quantification of plaque number for IMC43$^{AID}$, IMC43$^{AID}$ + IMC43$^{WT}$, IMC43$^{AID}$ + IMC43$^{\Delta1441-1561}$, and IMC43$^{AID}$ + IMC43$^{\Delta1562-1653}$ parasites after seven days of growth -/+ IAA. Y axis represents the number of plaques formed in +IAA conditions divided by the number of plaques formed in -IAA conditions for each strain. Statistical significance was determined by one-way ANOVA (****, P < 0.0001; ns = not significant). Scale bars = 2 μm.

enriched with a difference of >5 spectral counts when comparing the IMC43$^{TurboID}$ sample to the control sample (S1 Table). For the Y2H screen, we used Hybrigenics Services to screen against the *T. gondii* RH strain cDNA library using the full-length IMC43 protein as the bait.

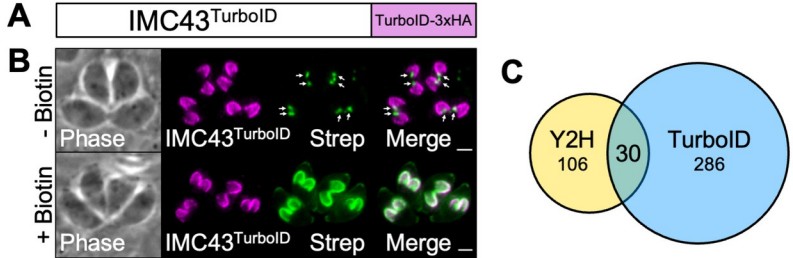

| Gene ID | Description | Y2H | TurboID Ctrl / Exp | GWCS | Localization |
|---|---|---|---|---|---|
| TGGT1_253440 | SRPK | B | 0 / 83.5 | -3.09 | nuclear (LOPIT) |
| TGGT1_297120 | hypothetical protein | B | 0 / 38 | -0.78 | N/A (LOPIT) |
| TGGT1_290170 | PPKL | C | 0 / 6 | -5.02 | multiple locations, daughter-enriched[54] |
| TGGT1_306060 | RON8 | C | 0 / 6 | -3.26 | rhoptries[76] |
| TGGT1_291180 | hypothetical protein | D | 0 / 165 | -4.69 | nuclear (LOPIT) |
| TGGT1_294610 | IMC30 | D | 1.5 / 163 | -0.98 | daughter IMC[29] |
| TGGT1_235340 | ISC1 | D | 1 / 155 | 0.29 | sutures[15] |
| TGGT1_203010 | Ark3 | D | 0 / 130.5 | -1.3 | centrosome & IMC-associated[55] |
| TGGT1_210345 | hypothetical protein | D | 0 / 91 | -1.79 | N/A (LOPIT) |
| TGGT1_291140 | CCR4-Not complex component, Not1 protein | D | 0 / 80 | -4.57 | nuclear (LOPIT) |
| TGGT1_231640 | IMC1 | D | 9 / 85 | -4 | mother & daughter IMC[13] |
| TGGT1_258540 | phosphoglycerate mutase family protein | D | 0 / 75 | -4.95 | nuclear & peripheral (LOPIT) |
| TGGT1_212770 | IMC34 | D | 0 / 57 | -1.8 | daughter IMC[48] |
| TGGT1_315580 | hypothetical protein | D | 0 / 56.5 | -0.67 | N/A (LOPIT) |
| TGGT1_223790 | hypothetical protein | D | 0 / 53.5 | 0 | apical (LOPIT) |
| TGGT1_269730 | myosin-light-chain kinase | D | 0 / 45 | 0.56 | N/A (LOPIT) |
| TGGT1_255450 | IMC31 | D | 0 / 25 | -1.78 | daughter IMC[29] |
| TGGT1_230000 | hypothetical protein | D | 0 / 20 | -5.29 | nuclear (LOPIT) |
| TGGT1_225690 | AC7 | D | 0 / 18.5 | -0.02 | apical cap[15] |
| TGGT1_230850 | TSC3 | D | 1 / 17.5 | -0.38 | sutures[16] |
| TGGT1_232150 | IMC32 | D | 0 / 15.5 | -4.32 | daughter IMC[27] |
| TGGT1_266640 | Acetyl-coenzyme A synthetase 2, putative | D | 0 / 15 | 0.59 | cytosol (LOPIT) |
| TGGT1_228750 | CDPK7 | D | 0 / 10 | -4.13 | cytosol[56] |
| TGGT1_214090 | signal peptidase | D | 0 / 9.5 | -0.3 | nuclear (LOPIT) |
| TGGT1_263000 | Beige/BEACH domain-containing protein | D | 0 / 8.5 | -0.94 | nuclear (LOPIT) |
| TGGT1_258410 | PHIL1 | D | 0 / 7 | 1.74 | IMC[49] |
| TGGT1_266830 | Sec7 domain-containing protein | D | 0 / 7 | 0.3 | nuclear & peripheral (LOPIT) |
| TGGT1_226510 | Sec23/Sec24 trunk domain-containing protein | D | 0 / 6.5 | -4.28 | nuclear (LOPIT) |
| TGGT1_246720 | hypothetical protein | D | 0 / 6 | 0.24 | conoid & daughter IMC[50] |
| TGGT1_210700 | hypothetical protein | D | 0 / 5.5 | -1.95 | nuclear (LOPIT) |

**Fig 5. TurboID and Y2H screens yield 30 candidate IMC43 binding partners.** A) Diagram of IMC43[TurboID]. B) IFA showing that IMC43[TurboID] localizes normally to the daughter IMC and biotinylates proximal proteins in a biotin-dependent manner. Arrows point to endogenously biotinylated apicoplasts. Magenta = anti-HA detecting IMC43[TurboID], Green = streptavidin. Scale bars = 2 μm. C) Venn diagram comparing genes identified in the Y2H screen and TurboID experiments. All Y2H hits that were ranked as "D" (moderate confidence) or higher were included. TurboID results were filtered to include only genes that were at least two-fold enriched with a difference of >5 spectral counts when comparing IMC43[TurboID] to control. 30 genes were identified in both experiments after filtering results as described. D) Table showing the 30 genes identified in both the TurboID and Y2H screens. The two candidate binding partners analyzed in this study are highlighted in yellow. "Y2H" column indicates the confidence score assigned in the Y2H screen (B = high confidence, C = good confidence, D = moderate confidence). "TurboID Ctrl / Exp" column indicates the average spectral count for each gene in the control and IMC43[TurboID] mass spectrometry results. "GWCS" refers to the phenotype score assigned to each gene in a genome-wide CRISPR/Cas9 screen [35]. "Localization" column reports the known localization of each protein [13,15,16,27,29,48–50,54–56,76]. Localizations followed by "(LOPIT)" indicate predicted localizations based on hyperplexed localization of organelle proteins by isotope tagging (hyperLOPIT) [42].

We included any hits that were assigned a confidence score of "A" (very high), "B" (high), "C" (good), or "D" (moderate) in our analysis (S2 Table). A total of 30 proteins were identified using both approaches (Fig 5C and 5D). To prioritize proteins for further analysis, we

considered their enrichment in the TurboID experiment, confidence score in the Y2H screen, transcriptional profile throughout the cell cycle, and known or predicted association with the daughter IMC [24,42]. Based on these criteria we selected two candidate IMC43 binding partners to explore further: the essential daughter protein IMC32 and the hypothetical protein encoded by TGGT1_297120.

## TGGT1_297120 encodes a novel IMC protein with a dynamic localization that is dependent on IMC43

TGGT1_297120 was designated as a "B" hit in our Y2H screen and was highly enriched in our IMC43[TurboID] results (Figs 5D and 6A). It also exhibits a cyclical transcriptional profile throughout the cell cycle, closely mirroring that of IMC43 (Fig 6B) [24]. Endogenous tagging of the gene in our IMC43[AID] parasites revealed that the protein is absent from mature parasites and exhibits a dynamic localization pattern in the IMC of daughter buds. At bud initiation, TGGT1_297120 appears in a small ring shape, similar to IMC43. During early budding, the protein appears to be restricted to the body of the daughter IMC. As the daughter buds expand, TGGT1_297120 maintains its localization in the body of the IMC and additionally becomes enriched in the basal complex. Finally, during late endodyogeny and bud maturation, the protein shifts completely to the basal complex, where it localizes just adjacent to MORN1, and finally disappears once the buds emerge as mature cells (Fig 6C and 6D). Given the dynamic localization of TGGT1_297120 within the daughter IMC, we named the protein IMC44.

To explore the function of IMC44, we used CRISPR/Cas9 to disrupt its gene, which was confirmed by IFA and PCR (S6A and S6B Fig). The Δ*imc44* parasites did not have any gross morphological defects (S6A Fig). Plaque assays showed that loss of IMC44 had no effect on overall lytic ability, as expected based on its modest phenotype score (S6C and S6D Fig). We next endogenously tagged IMC43 in the Δ*imc44* parasite line. IFA showed that the localization of IMC43 was unaffected in the Δ*imc44* parasites, indicating that IMC43 localizes independently of its partner IMC44 (S6E Fig). Together, these data indicate that IMC44 is dispensable for parasite fitness and does not play a role in IMC43 localization.

We next wanted to assess how depletion of IMC43 affects the localization of IMC44. When IMC43 was depleted, IMC44 failed to localize to the body of the IMC during the early stages of endodyogeny and instead localized to the basal complex (Fig 6E). Normal IMC44 localization was restored when we complemented with IMC43[WT], confirming that this change in localization was due to the loss of IMC43 (Fig 6F). To determine if this mislocalization could be linked to a specific region of IMC43, we endogenously tagged IMC44 in each of our ten IMC43 deletion constructs and used IFA to assess whether each deletion could rescue the mislocalization of IMC44 in the presence of IAA. Only the Δ1441–1561 mutant failed to rescue the mislocalization of IMC44, indicating that the essential region of IMC43 plays a role in the localization of IMC44 (Figs 6G and S7). To determine whether this region is directly involved in binding to IMC44, we used pairwise Y2H assays. Fragments of IMC43 and IMC44 were cloned into Y2H bait and prey vectors, co-transformed into yeast, and spotted on permissive (-L/W) and restrictive (-L/W/H) media to assess binding. These experiments demonstrated that the C-terminus of IMC43, which contains the essential domain, is sufficient for binding to IMC44 at either its N-terminus or its C-terminus (Fig 6H and 6I). Together these data indicate that the essential C-terminal domain of IMC43 binds to IMC44 and regulates its localization during endodyogeny.

## IMC43 regulates the organization of IMC32 in the daughter cell scaffold

The second candidate IMC43 binding partner we explored was IMC32, a daughter-enriched IMC body protein that we previously showed is essential for parasite replication and survival

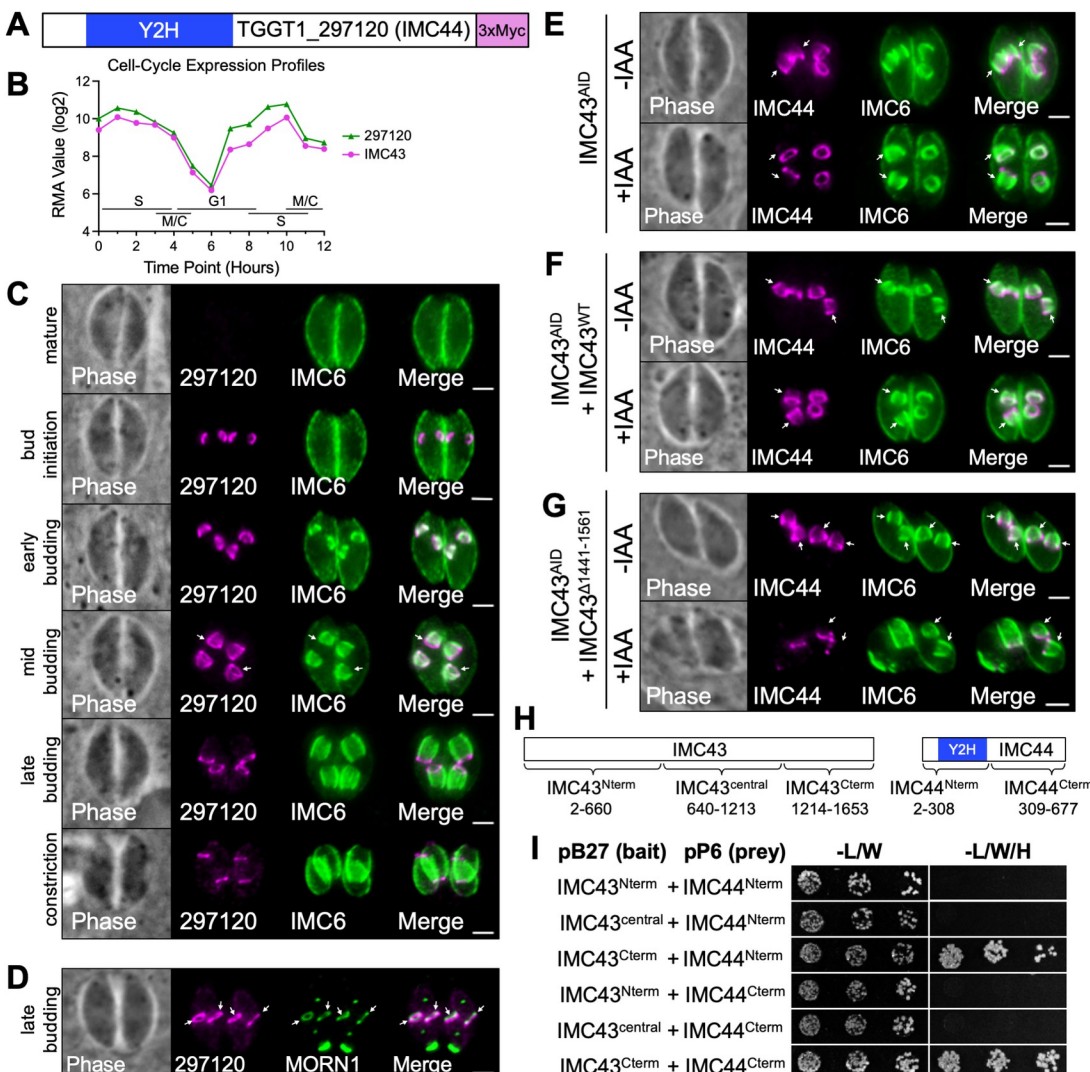

**Fig 6. IMC44 is a novel IMC protein which depends on IMC43 binding for regulation of its dynamic localization.** A) Gene model of TGGT1_297120 (IMC44) showing the region determined to interact with IMC43 by the Y2H screen. B) The cell-cycle expression profile for TGGT1_297120 closely mirrors the cyclical pattern of IMC43. RMA = robust multi-array average [24]. C) IFAs showing the localization of TGGT1_297120 at different stages of endodyogeny. Arrows indicate the point at which TGGT1_297120 can be first seen localizing to the basal complex of daughter buds. Magenta = anti-Myc detecting TGGT1_297120[3xMyc], Green = anti-IMC6. D) IFAs comparing the localization of TGGT1_297120 with the basal complex marker MORN1. Arrows point to the basal complex of daughter buds. Magenta = anti-Myc detecting TGGT1_297120[3xMyc], Green = anti-V5 detecting MORN1[3xV5]. E) IFA showing the localization of IMC44 in IMC43[AID] parasites -/+ IAA. Depletion of IMC43 causes IMC44 to fail to localize to the body of the IMC. Arrows point to the body of the IMC in daughter buds. Magenta = anti-Myc detecting IMC44[3xMyc], Green = anti-IMC6. F) IFA showing the localization of IMC44 in IMC43[AID] + IMC43[WT] parasites -/+ IAA. IMC43[WT] rescues the mislocalization of IMC44 observed in IMC43-depleted parasites. Arrows point to the body of the IMC in daughter buds. Magenta = anti-Myc detecting IMC44[3xMyc], Green = anti-IMC6. G) IFA showing the localization of IMC44 in IMC43[AID] + IMC43[Δ1441−1561] parasites -/+ IAA. IMC43[Δ1441−1561] fails to rescue the mislocalization of IMC44 observed in IMC43-depleted parasites. Arrows point to the body of the IMC in daughter buds. Magenta = anti-Myc detecting IMC44[3xMyc], Green = anti-IMC6. Scale bars = 2 μm. H) Diagrams showing the fragments of IMC43 and IMC44 that were fused to LexA/GAL4[AD] for pairwise Y2H assays. I) Pairwise Y2H assays assessing the interaction of IMC43 and IMC44. Growth on restrictive (-L/W/H) media indicates binding between the indicated fragments of each protein.

[27]. IMC32 contains five coiled-coil (CC) domains at its C-terminus, which we previously determined to be necessary for both its localization and function [27]. We also recently discovered a domain in the center of the protein which has structural homology to the fused Ig-PH

domain from the plant actin-binding protein SCAB1 (Fig 7A). IMC32 was identified as a "D" hit in our Y2H screen and was also relatively low abundance in our TurboID mass spectrometry results (Fig 5D). However, its localization, timing of recruitment, and function are all very similar to IMC43. Additionally, our Y2H screen data determined that three of the essential CC domains of IMC32 are involved in binding to IMC43, further supporting that IMC32 may be a true binding partner of IMC43 (Fig 7A).

To investigate the relationship between IMC32 and IMC43, we endogenously tagged IMC32 in our IMC43^AID line. As we previously reported, IMC32 localizes to five distinct puncta arranged in a pentagon during bud initiation. As the buds elongate, IMC32 extends into a series of five longitudinal stripes that stretch along the body of the daughter IMC (Fig 7B) [27]. When we examined IMC32 localization in IMC43-depleted parasites, we found that IMC32 exhibited normal localization during the bud initiation and early budding stages (Fig 7C). However, during mid- and late-budding, IMC32 became disorganized. The usual pattern of stripes along the body of the IMC was frequently missing, and instead, IMC32 had a diffuse staining pattern in the body of the daughter IMC and in the cytoplasm (Fig 7D). Accumulation of these defects during a longer IAA treatment led to increased cytoplasmic mislocalization (Fig 7E). Quantification showed that IMC32 was mislocalized in approximately 78% of vacuoles containing parasites at the mid- and late-budding stages when IMC43 was depleted. This defect was fully rescued by complementation with IMC43^WT (Fig 7F and 7G). This indicates that while IMC32 initially recruits to the DCS independently of IMC43, its' organization during later stages of endodyogeny is dependent on IMC43.

To determine whether IMC43 relies on IMC32 for its localization as well, we generated an IMC32^AID line in which IMC43 is endogenously tagged (S8A Fig). IFA showed efficient depletion of IMC32 resulting in severe morphological and replication defects as we have previously reported using an anhydrotetracycline transactivator-based regulation system (S8B Fig) [27]. IMC32 depletion had no effect on the localization of IMC43, indicating that IMC43 localizes independently of IMC32 (S8C Fig).

Next, we wanted to determine which region of IMC43 is required for the proper organization of IMC32 in the DCS. To assess this, we endogenously tagged IMC32 in each of our ten IMC43 deletion lines and used IFA to determine whether each deletion could rescue the mislocalization of IMC32. The Δ1441–1561 mutant was the only one that failed to rescue the mislocalization of IMC32, indicating that the essential region of IMC43 is involved in regulating the localization of IMC32 (Figs 7H and S9). To directly assess the binding interaction between IMC43 and IMC32, we once again used pairwise Y2H assays. These experiments demonstrated that the C-terminus of IMC43 is sufficient for binding to IMC32's CC domains (Fig 7I and 7J). Together, these data indicate that the essential C-terminal domain of IMC43 binds to IMC32's CC domains and controls its localization during the middle and late stages of endodyogeny.

## Discussion

In this study, we identified and characterized IMC43 as a novel daughter-specific IMC protein which plays an essential role in endodyogeny. We also used proximity labelling and Y2H approaches to identify and explore its binding partners. Our data demonstrates that depletion of IMC43 results in severe morphological and replication defects, resulting in a loss of overall lytic ability. The phenotype of IMC43-depleted parasites closely resembles that of IMC32-depleted parasites [27]. Loss of both proteins results in cytoskeletal instability, swelling, asynchronous division, formation of more than two daughter buds per cell, and excessive rounds of nuclear division and centrosome duplication (Fig 8A) [27]. The observed morphological defects are also similar to that observed for BCC0 [28]. One key difference between IMC43

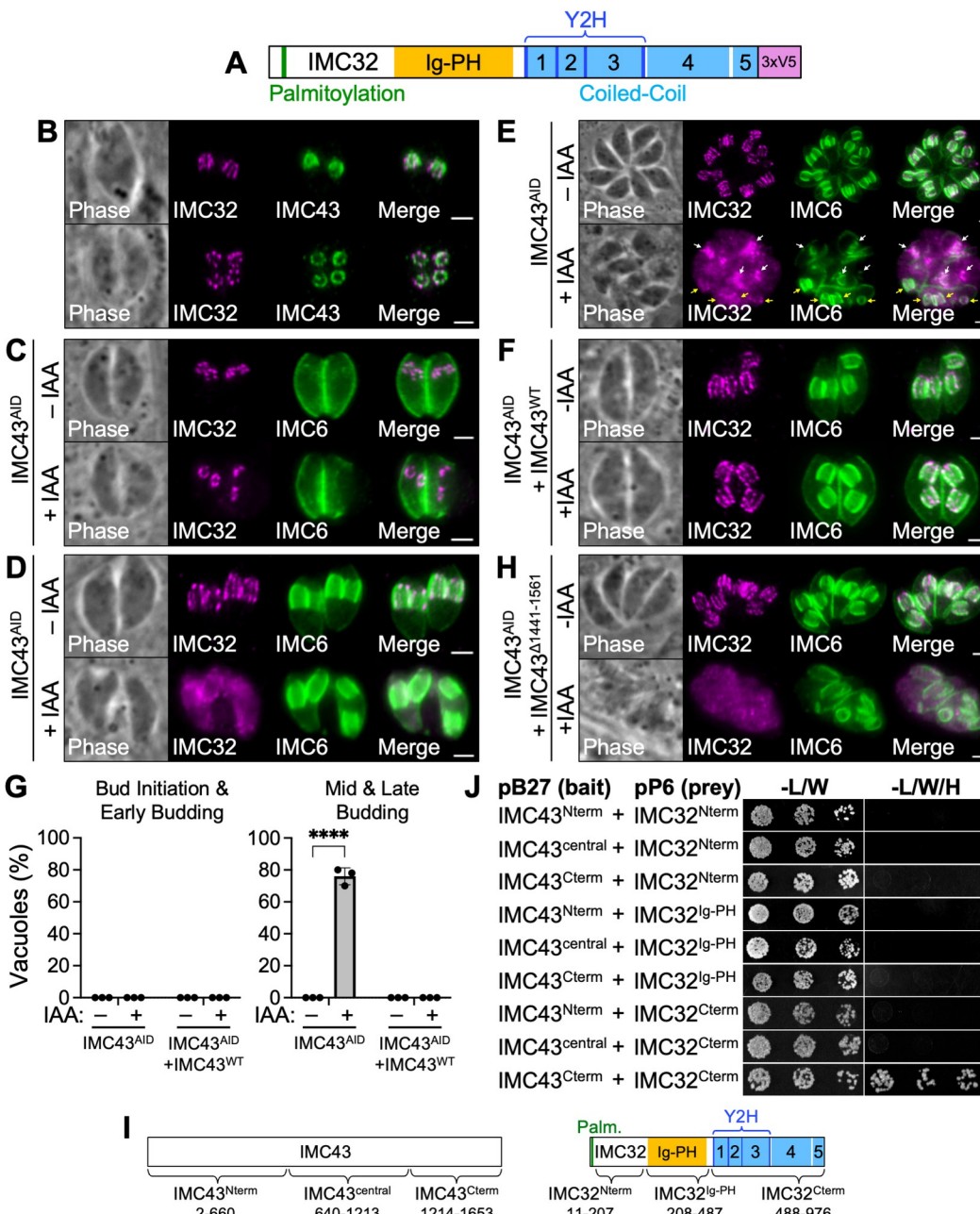

**Fig 7. IMC43 binding is required for maintenance of IMC32 during later stages of endodyogeny.** A) Gene model of IMC32 showing its predicted N-terminal palmitoylation site, Ig-PH domain, 5 predicted CC domains, and IMC43-interacting region identified by the Y2H screen. B) IFA comparing the localization of IMC43 and IMC32 at the mid/late budding stages of endodyogeny. Top: Daughter buds viewed from the side. Bottom: Daughter buds viewed from below. Magenta = anti-V5 detecting IMC32$^{3xV5}$, Green = anti-HA detecting IMC43$^{AID}$. C) IFA showing the localization of IMC32 during bud initiation in IMC43$^{AID}$ parasites after 18 hours of growth -/+ IAA. Depletion of IMC43 has no effect on IMC32 localization during bud initiation. Magenta = anti-V5 detecting IMC32$^{3xV5}$, Green = anti-IMC6. D) IFA showing the localization of IMC32 during the late-budding stage in IMC43$^{AID}$ parasites after 18 hours of growth -/+ IAA. Depletion of IMC43 causes IMC32 to lose its characteristic striped localization pattern at this stage of endodyogeny. Magenta = anti-V5 detecting IMC32$^{3xV5}$, Green = anti-IMC6. E) IFA showing the localization of IMC32 in IMC43$^{AID}$ parasites after 30 hours of growth -/+ IAA. Depletion of IMC43 over a longer time course results in an accumulation of mislocalized IMC32 in the cytoplasm. White arrows point to IMC32 enriching at early daughter buds. Yellow arrows point to daughter buds at the mid/late-budding stage with diffuse and disorganized IMC32 staining. Magenta = anti-V5 detecting IMC32$^{3xV5}$, Green = anti-IMC6. F) IFA showing the localization of IMC32 in IMC43$^{AID}$ + IMC43$^{WT}$ parasites after 18 hours of growth -/+ IAA. IMC43$^{WT}$ rescues the mislocalization of IMC32 caused by depletion of IMC43. Magenta = anti-V5 detecting

IMC32³ˣV⁵, Green = anti-IMC6. G) Quantification of vacuoles with mislocalized IMC32 at different stages of endodyogeny after 18 hours of growth -/+ IAA. Approximately 76% of vacuoles at the mid and late budding stages exhibited mislocalization of IMC32 when IMC43 was depleted. This phenotype was fully rescued by complementation with IMC43$^{WT}$. Statistical significance was determined using multiple two-tailed t tests (****, P < 0.0001; ns = not significant). H) IFA showing the localization of IMC32 in IMC43$^{AID}$ + IMC43$^{\Delta 1441-1561}$ parasites after 24 hours of growth -/+ IAA. IMC43$^{\Delta 1441-1561}$ fails to rescue the mislocalization of IMC32 caused by depletion of IMC43. Magenta = anti-V5 detecting IMC32$^{3xV5}$, Green = anti-IMC6. Scale bars = 2 μm. I) Diagrams showing the fragments of IMC43 and IMC32 that were fused to LexA/GAL4$^{AD}$ for pairwise Y2H assays. J) Pairwise Y2H assays assessing the interaction of IMC43 and IMC32. Growth on restrictive (-L/W/H) media indicates binding between the indicated fragments of each protein.

and BCC0 is that loss of BCC0 was reported to cause a partial disruption of the basal complex in daughter buds, whereas we demonstrated that the basal complex is unaffected by loss of IMC43. Finally, the severe replication defects are also similar to Δ*imc29* parasites, although these defects are observed at a higher frequency in IMC43-depleted parasites [29].

While IMC43 does not contain any functional domains identifiable by sequence or structural homology, our deletion analyses identified two important regions. We first identified a small region towards the N-terminus (residues 402–494) which plays a minor role in localization of the protein. Deletion of this region led to a partial mislocalization of the protein to the cytoplasm. While this partial mislocalization could be due to protein misfolding, the fact that the Δ402–494 mutant rescues the phenotype of the knockdown makes this less likely.

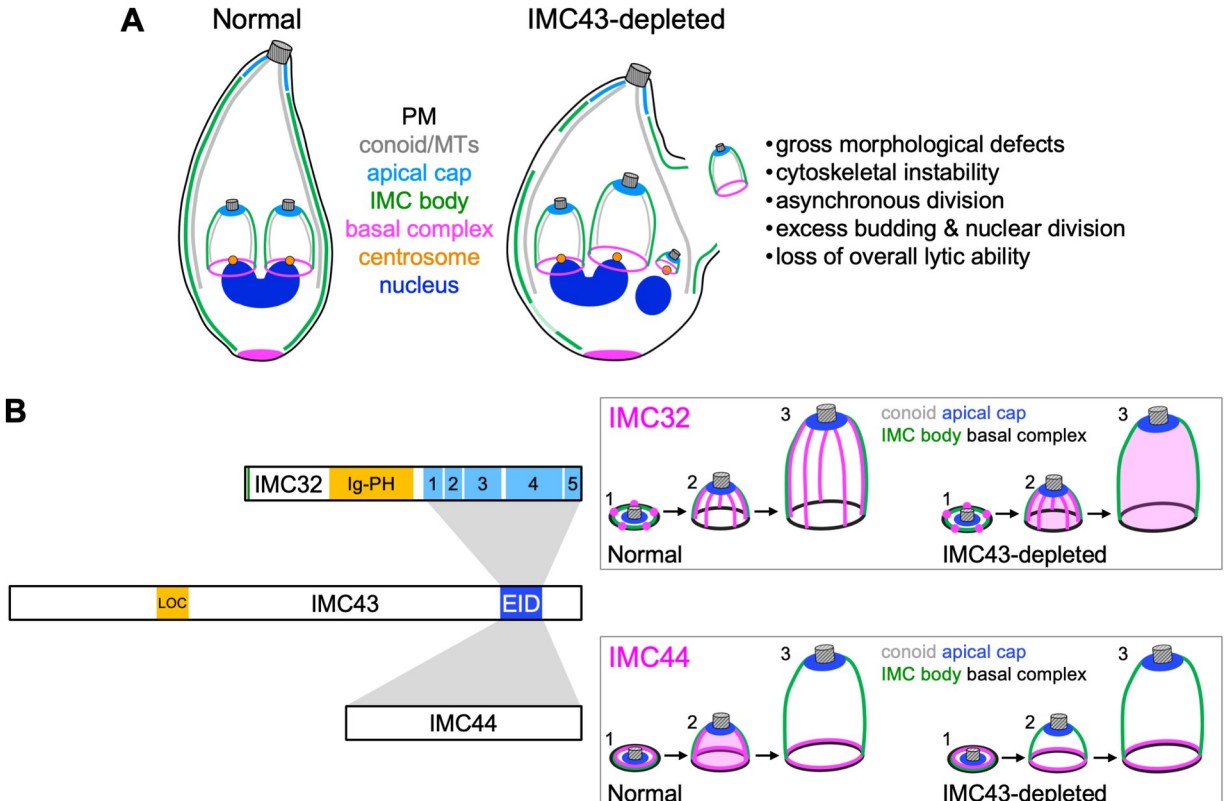

**Fig 8. Summary.** A) Diagram summarizing the effects of IMC43 depletion. Depletion of IMC43 causes gross morphological defects such as swelling, cytoskeletal instability, asynchronous division, excess budding & nuclear division, and loss of overall lytic ability. Assembly of the basal complex and apical cap remain unaffected. B) Diagram summarizing the relationship between IMC43 and its binding partners IMC32 and IMC44. The newly identified Localization domain (LOC) and Essential Interaction Domain (EID) of IMC43 are shown. Grey boxes indicate interactions between the Essential Interaction Domain of IMC43 and IMC32/IMC44. Boxed panels include a diagram showing how depletion of IMC43 affects the localization of IMC32 and IMC44 throughout bud initiation (1), early/mid-budding (2), and late budding (3).

Furthermore, this region of the protein is more conserved than nearby regions, supporting the hypothesis that this region may contain a domain involved in localization. Given these data, we tentatively assign the region encompassed by residues 402–494 as the Localization Domain (LOC) (Fig 8B). However, since none of the deletions fully abolished targeting to the IMC, other regions of the protein must play a role in localization. We also identified a domain near the C-terminus of the protein which is essential for the function of IMC43. This region is highly conserved and is one of the few regions of the protein that is not predicted to be intrinsically disordered [43]. AlphaFold predicts a four-helix bundle in this region, although the confidence is relatively weak [33,44,45]. Four-helix bundles are structurally similar to coiled-coils which often play roles in protein-protein interactions [46,47]. Based on its essential function in facilitating interaction between IMC43 and its partners, we named this region the Essential Interaction Domain (EID) (Fig 8B).

By combining the results of TurboID proximity labelling and a Y2H screen, we identified 30 candidate IMC43 binding partners and investigated two of them in relation to IMC43. This led us to two key findings: the identification of IMC44 as a novel daughter IMC protein with a dynamic localization which is dependent on IMC43, and the discovery of an essential IMC43-IMC32 daughter bud assembly complex that is critical for endodyogeny. IMC44 was found to have a dynamic localization pattern in which it localizes to the body of the IMC during the early stages of endodyogeny and then transitions to the basal complex during the middle and late stages. A similar dynamic localization has previously been reported for the alveolins IMC5, -8, -9, and -13, and also for the recently discovered BCC3 [14,28]. IMC44's localization within the IMC body is similar to the alveolins, whereas BCC3 appears to be present in the sutures of the daughter bud. However, the timing of IMC44's shift to the basal complex is more like that of BCC3, which also begins to transition to the basal complex during mid-budding, rather than just prior to bud constriction like the alveolins. Interestingly, all five of these proteins, as well as IMC44, have moderate fitness scores, and we showed here that IMC44 is dispensable [35]. Given their shared localization patterns, it's possible that there may be some functional redundancy among these proteins that explains their dispensability. For IMC44, we found that localization to the IMC body is dependent on IMC43, while localization to the basal complex occurs independent of IMC43. It's possible that IMC44's localization to the basal complex relies instead on binding to another basal complex-localizing protein. Since IMC5, -8, -9, and -13 shift to the basal complex later than IMC44, it will be interesting to see whether disruption of IMC44 affects any of their localizations.

The identification of IMC32 as an IMC43 binding partner was exciting due to their similar essential functions in endodyogeny. Intriguingly, we found that while IMC32 can localize to the early DCS during bud initiation independently, the maintenance of its striped localization pattern during mid and late budding requires IMC43. We also demonstrated that the C-terminus of IMC43 is sufficient for binding to IMC32's C-terminal CC domains (Fig 8B). In our previous study, we showed that the localization of IMC32 requires both its predicted N-terminal palmitoylation site and at least one of its C-terminal CC domains [27]. This suggests a model in which IMC32 initially recruits to the DCS via palmitoylation and is maintained in the later stages of endodyogeny by binding to IMC43. Together, our data indicates that IMC43 and IMC32 make up an essential protein complex in the DCS and are required for proper progression of endodyogeny. However, it is somewhat puzzling that IMC32 appears to be conserved throughout the Apicomplexa, whereas clear orthologs of IMC43 are only found in *N. caninum*, *Hammondia hammondi*, and *Besnoitia besnoiti* [27]. One possibility is that an ortholog of IMC43 is present in other apicomplexans but has diverged so much that it's no longer recognizable by sequence homology. This hypothesis is supported by the fact that the 43 kDa *Sarcocystic neurona* protein encoded by SN3_03500255 and the 62 kDa *Cystoisospora suis*

protein encoded by CSUI_009138 have significant homology to the essential C-terminus of IMC43. Alternatively, it is possible that IMC32 evolved to localize independently of other proteins in other apicomplexan parasites or relies on a different binding partner. Investigating the function of IMC32 in *Plasmodium spp.* would provide more insight into this question.

Of the remaining 28 candidate IMC43 binding partners, 25 were identified as "D" hits in the Y2H screen. "D" hits indicate moderate confidence in the interaction and represent a mix of false-positives and genuine interactions that are difficult to detect by Y2H due to low mRNA abundance in the *T. gondii* library, difficulty folding in yeast, or toxicity when expressed in yeast [40,41]. Given this caveat, it's important to directly interrogate each candidate to determine whether it's likely to be a bona fide binding partner. One example of this is IMC32, which was scored as a D hit and was validated in this study as a true partner of IMC43. It's likely that IMC32 had a moderate confidence score due to low mRNA abundance in the *T. gondii* library, which was produced using extracellular (non-budding) parasites. Other D hits include several proteins which are known to localize to daughter buds, including IMC30, IMC31, IMC34, PHIL1, IMC1, and TGGT1_246720 [13,29,48–50]. Given their shared localization with IMC43, these proteins represent interesting targets for future studies.

Another interesting finding from our interaction screening experiments was the identification of three kinases and one phosphatase as candidate IMC43 binding partners. Phosphoproteomics studies have identified several phosphorylation sites in IMC43, raising the possibility that phosphorylation could be important for the regulation of IMC43 localization or function. The putative cell-cycle-associated protein kinase SRPK was identified as a B hit and was found to bind via its kinase domain to IMC43 [51]. Similarly, the kelch repeat domain-containing serine/threonine protein phosphatase protein PPKL was identified as a "C" hit and was found to bind to IMC43 at two locations: its phosphatase domain and a region just N-terminal to the kelch repeat domain [52]. The fact that IMC43 binds to these proteins at their active sites suggests that IMC43 could be a substrate of SRPK and/or PPKL. SRPK has not been studied in *T. gondii*, but its low phenotype score suggests an essential function [35]. In *P. falciparum*, PPKL has been shown to be crucial for ookinete development and microtubule tethering to the IMC, and a recent pre-print by Yang et al. demonstrated that it plays an essential role in regulating daughter bud formation in *T. gondii* [53,54]. Additionally, both the aurora kinase Ark3 and the calcium-dependent protein kinase CDPK7 were identified as "D" hits and were previously shown to be involved in parasite replication [55,56]. Neither protein was found to bind to IMC43 at their kinase domain, so they are less likely to phosphorylate IMC43. However, IMC43 could still bind to these proteins to regulate their localization or activity. Exploring these candidate interactors and determining how their activity affects the function of IMC43 will be a fascinating topic of future studies.

Together, the data presented in this study identifies IMC43 as a foundational protein involved in *T. gondii* endodyogeny and reveals the existence of an essential IMC43-IMC32 daughter bud assembly complex. The protein-protein interaction data presented in this study yields ample opportunities for future studies that will inform our knowledge of how daughter IMC assembly is regulated. Furthering our understanding of this process will be vital for the identification of novel drug and vaccine targets for diseases caused by apicomplexan parasites.

## Materials and methods

### *T. gondii* and host cell culture

Parental *T. gondii* RH*ΔhxgprtΔku80* (wild-type) and subsequent strains were grown on confluent monolayers of human foreskin fibroblasts (BJ, ATCC, Manassas, VA) at 37°C and 5% $CO_2$ in Dulbecco's Modified Eagle Medium (DMEM) supplemented with 5% fetal bovine serum (Gibco), 5% Cosmic calf serum (Hyclone), and 1x penicillin-streptomycin-L-glutamine

(Gibco). Constructs containing selectable markers were selected using 1 μM pyrimethamine (dihydrofolate reductase-thymidylate synthase [DHFR-TS]), 50 μg/mL mycophenolic acid-xanthine (HXGPRT), or 40 μM chloramphenicol (CAT) [57–59]. Homologous recombination to the UPRT locus was negatively selected using 5 μM 5-fluorodeoxyuridine (FUDR) [60].

## Antibodies

The HA epitope was detected with mouse monoclonal antibody (mAb) HA.11 (BioLegend; 901515) and rabbit polyclonal antibody (pAb) anti-HA (Invitrogen; PI715500). The Ty1 epitope was detected with mouse mAb BB2 [61]. The c-Myc epitope was detected with mouse mAb 9E10 [62]. The V5 epitope was detected with mouse mAb anti-V5 (Invitrogen; R96025) and rabbit mAb anti-V5 (Cell Signaling Technology; 13202S). The OLLAS epitope was detected with rat mAb anti-OLLAS [63]. *Toxoplasma*-specific antibodies include rabbit pAb anti-IMC6 [64], mouse mAb anti-IMC1 [65], mouse mAb anti-ISP1 [32], rabbit pAb anti-Centrin1 (Kerafast; EBC004) [66], and rabbit anti-Catalase [67].

## Endogenous epitope tagging and knockout

For C-terminal endogenous tagging, a pU6-Universal plasmid containing a protospacer against the 3′ untranslated region (UTR) of the target protein approximately 100 bp downstream of the stop codon was generated, as described previously [68]. A homology-directed repair (HDR) template was PCR amplified using the Δ*ku80*-dependent LIC vector pmAID3xHA.LIC-HXGPRT, pmAID3xTy.LIC-HXGPRT, p3xHA.LIC-DHFR, p3xMyc. LIC-DHFR, p2xStrep3xTy.LIC-CAT, p2xStrep3xTy.LIC-HXGPRT, p3xV5.LIC-DHFR, p3xV5.LIC-HXGPRT, psmOLLAS.LIC-DHFR, or pTurboID3xHA.LIC-DHFR, all of which include the epitope tag, 3′ UTR, and a selection cassette [69]. The 60-bp primers include 40 bp of homology immediately upstream of the stop codon or 40 bp of homology within the 3′ UTR downstream of the CRISPR/Cas9 cut site. Primers that were used for the pU6-Universal plasmid as well as the HDR template are listed in S3 Table.

For N-terminal endogenous tagging of MORN1, a synthetic gene containing 146 bp of homology upstream of the start codon, a 3xV5 tag, and 219 bp of homology downstream of the start codon (primer P55) was ligated into a pJet vector (MORN1.3xV5.pJet). A homology-directed repair (HDR) template was PCR amplified from the MORN1.3xV5.pJet vector using primers P58 and P59. The protospacer was designed to target the sequence just downstream of the MORN1 start codon and was ligated into the pU6-Universal plasmid.

For knockout of IMC43 and IMC44, the protospacer was designed to target the coding region of the gene of interest, ligated into the pU6-Universal plasmid and prepared similarly to the endogenous tagging constructs. The HDR template was PCR amplified from a pJET vector containing the HXGPRT drug marker driven by the NcGRA7 promoter using primers that included 40 bp of homology immediately upstream of the start codon or 40 bp of homology downstream of the region used for homologous recombination for endogenous tagging.

For all tagging and knockout constructs, approximately 50 μg of the sequence-verified pU6-Universal plasmid and the PCR-amplified HDR template were electroporated into the appropriate parasite strain. Transfected cells were allowed to invade a confluent monolayer of HFFs overnight, and appropriate selection was applied. Successful tagging was confirmed by IFA, and clonal lines of tagged parasites were obtained through limiting dilution.

## Complementation of IMC43^AID parasites

The IMC43 coding region was PCR amplified from cDNA using primers P9 and P10. This was cloned into pUPRTKO-IMC32HA [27] using BglII/NotI to make pUPRTKO-IMC32p-

IMC43-3xHA. Next, the endogenous promoter (EP) was amplified from genomic DNA using primers P11 and P12 and the entire pUPRTKO-IMC32p-IMC43-3xHA plasmid except for the IMC32 promoter was amplified using primer P13 and P14. The two fragments were then ligated using Gibson Assembly to make pUPRTKO-EP-IMC43-3xHA (E2611, NEB). Finally, an smOLLAS tag was amplified from the p-smOLLAS-DHFR-LIC vector using primers P15 and P16 and the entire pUPRTKO-EP-IMC43-3xHA plasmid except for the 3xHA tag was amplified using primers P17 and P18. The two fragments were then ligated using Gibson assembly to make pUPRTKO-EP-IMC43$^{WT}$-smOLLAS (IMC43$^{WT}$). This complement vector was then linearized with PsiI-HFv2 and transfected into IMC43$^{AID}$ parasites along with a pU6 that targets the UPRT coding region. Selection was performed with 5 µg/mL 5-fluorodeoxyuridine (FUDR) for replacement of UPRT. Clones were screened by IFA, and an smOLLAS-positive clone was designated IMC43$^{WT}$. IMC43$^{WT}$ was used as the template to generate deletion constructs using Q5 site-directed mutagenesis with primers P19-P38 (E0552S, NEB). The same processes for linearization, transfection, and selection were followed for all deletion and mutant constructs. All restriction enzymes were purchased from NEB.

## Immunofluorescence assay

Confluent HFF cells were grown on glass coverslips and infected with *T. gondii*. After 18–40 hours, the coverslips were fixed with 3.7% formaldehyde in PBS and processed for immunofluorescence as described [70]. Primary antibodies were detected by species-specific secondary antibodies conjugated to Alexa Fluor 594/488/405 (ThermoFisher). Coverslips were mounted in Vectashield (Vector Labs), viewed with an Axio Imager.Z1 fluorescent microscope and processed with ZEN 2.3 software (Zeiss).

## Western blot

Parasites were lysed in 1x Laemmli sample buffer with 100 mM DTT and boiled at 100˚C for 5 minutes. Lysates were resolved by SDS-PAGE and transferred to nitrocellulose membranes, and proteins were detected with the appropriate primary antibody and corresponding secondary antibody conjugated to horseradish peroxidase. Chemiluminescence was induced using the SuperSignal West Pico substrate (Pierce) and imaged on a ChemiDoc XRS+ (Bio-Rad).

## Plaque assay

HFF monolayers were infected with 200 parasites/well of individual strains and allowed to form plaques for 7 days. Cells were then fixed with ice-cold methanol and stained with crystal violet. All plaque assays were performed in triplicate. To quantify plaque size, the areas of 50 plaques per condition were measured using ZEN software (Zeiss). To quantify plaque number, the total number of plaques in each condition was counted manually. Graphical and statistical analyses were performed using Prism GraphPad 8.0.

## Quantification of defective vacuoles and mislocalized IMC32

For quantification of morphological and replication defects, HFF monolayers grown on glass coverslips were infected with IMC43$^{AID}$ parasites at low MOI -/+ IAA. 24 hours post-infection, coverslips were fixed with 3.7% PFA, processed for immunofluorescence, and labeled with anti-HA and anti-IMC6. For both the -IAA and +IAA conditions, >100 vacuoles spread across at least 10 fields were scored as defective or normal. Vacuoles were scored as defective if they exhibited any of the following defects: desynchronized endodyogeny, >2 daughter buds per cell, immature daughter cells present outside of a maternal cell, gaps in the cytoskeleton

marked by IMC6, or severe swelling. The experiment was performed in triplicate. Significance was determined using a two-tailed t-test.

For quantification of mislocalized IMC32, HFF monolayers grown on glass coverslips were infected with IMC32$^{3xV5}$IMC43$^{AID}$ parasites at low MOI -/+ IAA. 18 hours post-infection, coverslips were fixed with 3.7% PFA, processed for immunofluorescence, and labeled with anti-V5 and anti-IMC6. For both the -IAA and +IAA conditions, >100 vacuoles containing parasites at the bud initiation and early budding stages of endodyogeny spread across at least 10 fields were scored as either normal or mislocalized. Vacuoles were scored as mislocalized at these stages if IMC32 signal was not observed in 5 distinct puncta/short stripes arranged in a ring-shape on nascent daughter buds. Then, >100 different vacuoles containing parasites at the mid and late budding stages of endodyogeny were scored as either normal or mislocalized. Vacuoles were scored as mislocalized at these stages if IMC32 signal was not observed in a series of longitudinal stripes along the body of the IMC. The experiment was performed in triplicate. Significance was determined using a two-tailed t-test.

## Affinity capture of biotinylated proteins

For affinity capture of proteins from whole cell lysates, HFF monolayers infected with IMC43$^{TurboID}$ or control parasites (RHΔ*hxgprt*Δ*ku80*, WT) were grown in medium containing 150 μM biotin for 30 hours. Intracellular parasites in large vacuoles were collected by manual scraping, washed in PBS, and lysed in radioimmunoprecipitation assay (RIPA) buffer (50 mM Tris [pH 7.5], 150 mM NaCl, 0.1% SDS, 0.5% sodium deoxycholate, 1% NP-40) supplemented with Complete Protease Inhibitor Cocktail (Roche) for 30 min on ice. Lysates were centrifuged for 15 min at 14,000 x g to pellet insoluble material, and the supernatant was incubated with Streptavidin Plus UltraLink resin (Pierce) overnight at 4°C under gentle agitation. Beads were collected and washed five times in RIPA buffer, followed by three washes in 8 M urea buffer (50 mM Tris-HCl [pH 7.4], 150 mM NaCl) [71]. Samples were submitted for on-bead digests and subsequently analyzed by mass spectrometry. The experiment was performed in duplicate.

## Mass spectrometry of biotinylated proteins

Purified proteins bound to streptavidin beads were reduced, alkylated, and digested by sequential addition of Lys-C and trypsin proteases. Samples were then desalted using C18 tips (Pierce) and fractionated online using a 75-μm inner-diameter fritted fused silica capillary column with a 5-μm pulled electrospray tip and packed in-house with 25 cm of C18 (Dr. Maisch GmbH) 1.9-μm reversed-phase particles. The gradient was delivered by a 140-minute gradient of increasing acetonitrile and eluted directly into a Thermo Orbitrap Fusion Lumos instrument where MS/MS spectra were acquired by Data Dependent Acquisition (DDA). Data analysis was performed using ProLuCID and DTASelect2 implemented in Integrated Proteomics Pipeline IP2 (Integrated Proteomics Applications) [72–74]. Database searching was performed using a FASTA protein database containing *T. gondii* GT1-translated open reading frames downloaded from ToxoDB. Protein and peptide identifications were filtered using DTASelect and required a minimum of two unique peptides per protein and a peptide-level false positive rate of less than 5% as estimated by a decoy database strategy. Candidates were ranked by spectral count comparing IMC43$^{TurboID}$ versus control samples [75].

## Yeast two-hybrid

Y2H screening was performed by Hybrigenics Services as previously described [40,41]. Briefly, the full-length coding sequence of IMC43 was cloned into the pB35 vector (N-GAL4-bait-C fusion, inducible) and transformed in yeast. This construct was screened for interactions

against the *T. gondii* RH strain cDNA library with 37 million interactions tested. Confidence for each interaction was assessed algorithmically (Predicted Biological Score, PBS).

For pairwise Y2H assays, fragments of IMC43, IMC44, and IMC32 were cloned into the pB27 (N-LexA-bait-C fusion) or pP6 (N-GAL4$^{AD}$-prey-C fusion) vectors (Hybrigenics Services) as N-terminal fusions with the LexA DNA binding domain or GAL4 activation domain, respectively. All pB27 and pP6 constructs were cloned by Gibson Assembly using primers P68-P101. Pairs of pB27 and pP6 constructs were co-transformed into the L40 strain of *S. cerevisiae* [MATa his3D200trp1-901 leu2-3112 ade2 LYS2::(4lexAop-HIS3) URA3::(8lexAop-lacZ) GAL4]. Strains were grown overnight in permissive (-Leu/-Trp) medium, normalized to $OD_{600} = 2$, then spotted in six serial dilutions onto permissive (-Leu/-Trp) and restrictive (-Leu/-Trp/-His) media. Growth was assessed after 3–5 days.

## Supporting information

**S1 Fig. Conservation and secondary structure predictions for TGGT1_238895.** A) ToxoDB reports TGGT1_238895 as a 1,654 amino acid protein, with the first two residues being methionine. Alignment of TGGT1_238895 with its *N. caninum* homolog NCLIV_015750 show that the first methionine (lowercase, bold) in TGGT1_238895 is not conserved. The second methionine (uppercase, bold) and much of the N-terminus is highly conserved. The second methionine is also a more favorable start codon based on the *T. gondii* consensus translation initiation sequence (A at position -3, G at position +4) [77]. We therefore determined that the protein is likely to start at the second methionine, resulting in TGGT1_238895 encoding a 1,653 amino acid protein. Residues are numbered accordingly in this study. B) The amino acid sequence of IMC43 (TGGT1_238895) was aligned to its *N. caninum* ortholog NCLIV_015750 using ClustalO 1.2.4. Alpha-helices predicted by PSIPRED are shown above their corresponding sequences. Regions chosen for the deletion series are highlighted.
(TIF)

**S2 Fig. IFA showing successful targeting of the IMC43 locus using CRISPR/Cas9.** A) Diagram showing the strategy used to genetically disrupt IMC43 in an IMC43$^{3xHA}$ parental line. B) IFA showing that FLAG-positive/HA-negative parasites with severe morphological defects were observed 30 hours after transfection, indicating successful ablation of the target gene. The Δ*imc43* parasites were rapidly lost from the population and could not be recovered. Arrows point to daughter buds lacking HA staining. Magenta = anti-HA, Green = anti-FLAG detecting Cas9$^{3xFLAG}$, Blue = anti-IMC6. Scale bar = 2 μm.
(TIF)

**S3 Fig. Western blots showing depletion and complementation of IMC43$^{AID}$.** A) Western blot showing efficient depletion of IMC43$^{AID}$ after four hours of IAA treatment. Parasites were grown intracellularly for 24 hours prior to adding IAA. Catalase is used as a loading control. B) Western blot of IMC43$^{AID}$ + IMC43$^{WT}$ parasites after four hours of IAA treatment. IMC43$^{WT}$ expresses at equal levels -/+ IAA. Catalase is used as a loading control. C) Western blots comparing expression levels of IMC43$^{WT}$ with the IMC43 deletion constructs shown in [Fig 4A]. Catalase was used as a loading control. D) Western blots comparing expression levels of IMC43$^{WT}$ with IMC43$^{Δ1441-1561}$ and IMC43$^{Δ1562-1653}$. Catalase was used as a loading control.
(TIF)

**S4 Fig. Most IMC43 deletion constructs localize normally to the daughter IMC.** IFAs showing that eight of the IMC43 deletion constructs colocalize with IMC43$^{AID}$. Magenta = anti-OLLAS detecting IMC43 deletion constructs, Green = anti-HA detecting IMC43$^{AID}$. Scale bars = 2 μm.
(TIF)

**S5 Fig. Most IMC43 deletion constructs rescue the morphological and replication defects observed upon depletion of IMC43.** IFAs showing that eight of the IMC43 deletion constructs fully rescue the morphological and replication defects caused by depletion of IMC43. Magenta = anti-OLLAS detecting IMC43 deletion constructs, Green = anti-IMC6. Scale bars = 2 μm.
(TIF)

**S6 Fig. IMC44 is dispensable for the *T. gondii* lytic cycle and does not impact IMC43 localization.** A) The endogenous locus for IMC44 was disrupted in the IMC44$^{3xMyc}$ parent strain. IFA of Δ*imc44* parasites confirms loss of IMC44$^{3xMyc}$ signal. Magenta = anti-Myc, Green = anti-IMC6. B) PCR verification for genomic DNA of WT and Δ*imc44* parasites. Diagram indicates the binding location of primers used to amplify the IMC44 coding sequencing (blue arrows) and the site of recombination for the knockout (red arrows). C) Plaque assays of WT and Δ*imc44* parasites. D) Quantification of plaque size for plaque assays shown in panel D. Statistical significance was determined using a two-tailed t test (ns = not significant). E) IFA showing normal localization of IMC43 in Δ*imc44* parasites. Magenta = anti-Ty detecting IMC43$^{2xStrep3xTy}$, Green = anti-IMC6. Scale bars = 2 μm.
(TIF)

**S7 Fig. Most IMC43 deletion constructs rescue the mislocalization of IMC44 caused by depletion of IMC43.** IFAs showing the localization of IMC44 in nine of the IMC43 deletion lines. All nine shown in this figure rescue the mislocalization of IMC44. Magenta = anti-Myc detecting IMC44$^{3xMyc}$, Green = anti-IMC6. Scale bars = 2 μm.
(TIF)

**S8 Fig. Depletion of IMC32 has no effect on IMC43 localization.** A) An mAID-3xTy degron tag was fused to the C-terminus of IMC32 in a TIR1-expressing strain to facilitate proteasomal degradation upon treatment with IAA. B) IFA of IMC32$^{AID}$ parasites after 24 hours of growth -/+ IAA. Depletion of IMC32 results in morphological and replication defects as previously described [27]. Magenta = anti-Ty detecting IMC32$^{AID}$, Green = anti-IMC6. C) IFA showing the localization of IMC43 in IMC32$^{AID}$ parasites after 24 hours of growth -/+ IAA. IMC43 is unaffected by depletion of IMC32. Magenta = anti-HA detecting IMC43$^{3xHA}$, Green = anti-IMC6. Scale bars = 2 μm.
(TIF)

**S9 Fig. Most IMC43 deletion constructs rescue the mislocalization of IMC32 caused by depletion of IMC43.** IFAs showing the localization of IMC32 in nine of the IMC43 deletion lines. All nine shown in this figure rescue the mislocalization of IMC32. Magenta = anti-V5 detecting IMC32$^{3xV5}$, Green = anti-IMC6. Scale bars = 2 μm.
(TIF)

**S1 Table. Full IMC43$^{TurboID}$ mass spectrometry results.** Full list of genes identified by mass spectrometry in the IMC43$^{TurboID}$ experiment. Spectral counts are shown for each gene. "Enrichment Diff" refers to the difference between the average spectral count in IMC43$^{TurboID}$ and control parasites. "Enrichment Fold" refers to the average spectral count for IMC43$^{TurboID}$ samples divided by the average spectral count for control samples.
(XLSX)

**S2 Table. Full IMC43 Y2H screen results.** Full list of genes identified by the Hybrigenics Y2H screen. All clones identified for a specific gene are grouped. Clones that were found to be out-of-frame are greyed out. Global PBS indicates the confidence score assigned to each clone

[40,41].
(XLSX)

**S3 Table. Oligonucleotides used in this study.**
(XLSX)

## Acknowledgments

We thank Dominique Soldati-Favre for the catalase antibody, Gary Ward for the IMC1 antibody, and Michael Reese for the pP6 and pB27 plasmids, L40 strain of yeast, and protocols for pairwise Y2H assays. We thank members of the Bradley lab for helpful reading of the manuscript.

## Author Contributions

**Conceptualization:** Rebecca R. Pasquarelli, Peter J. Bradley.

**Data curation:** Jihui Sha, James A. Wohlschlegel.

**Formal analysis:** Rebecca R. Pasquarelli.

**Funding acquisition:** James A. Wohlschlegel, Peter J. Bradley.

**Investigation:** Rebecca R. Pasquarelli, Peter S. Back, Jihui Sha.

**Project administration:** James A. Wohlschlegel, Peter J. Bradley.

**Resources:** James A. Wohlschlegel, Peter J. Bradley.

**Supervision:** James A. Wohlschlegel, Peter J. Bradley.

**Validation:** Rebecca R. Pasquarelli.

**Visualization:** Rebecca R. Pasquarelli.

**Writing – original draft:** Rebecca R. Pasquarelli, Jihui Sha, Peter J. Bradley.

**Writing – review & editing:** Rebecca R. Pasquarelli, Peter J. Bradley.

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
