## [Decision Letter · Decision Letter 0]

23 Aug 2023

Dear Prof Bradley,

Thank you very much for submitting your manuscript "Identification of IMC43, a novel IMC protein that collaborates with IMC32 to form an essential daughter bud initiation complex in Toxoplasma gondii" for consideration at PLOS Pathogens. As with all papers reviewed by the journal, your manuscript was reviewed by members of the editorial board and by several independent reviewers. The reviewers appreciated the attention to an important topic. Based on the reviews, we are likely to accept this manuscript for publication, providing that you modify the manuscript according to the review recommendations.

I am returning your manuscript with three reviews. The reviewers were very positive about the quality and impact of the work of the manuscript. There are some minor aspects that I would like to you to address to prepare the manuscript for publication. No new experiments are required.

In particular, I'd like you to consider reviewer 2's points:

1. Re: the title of your manuscript. Please revise the title or provide justification as to why you think it should not be.

2. Please justify why you used the time point of 24hrs of IAA treatment given the fast knockdown and consider including this justification into the manuscript.

3. You are not required to perform additional IFAs looking at the location of the suture proteins, ISC and TSCs, unless you would like to.

4. No super resolution or additional experimentation to dissect out localisation of IMC43 truncation mutants are required.

Please also address point-by-point all other queries and modify the text of the manuscript accordingly.

If all the following items are addressed, I hope to be able to make a final decision without sending the manuscript out for a second round of review.

Sincerely,

Christopher J. Tonkin

Guest Editor

PLOS Pathogens

Dominique Soldati-Favre

Section Editor

PLOS Pathogens

Kasturi Haldar

Editor-in-Chief

PLOS Pathogens

orcid.org/0000-0001-5065-158X

Michael Malim

Editor-in-Chief

PLOS Pathogens

orcid.org/0000-0002-7699-2064

I am returning your manuscript with three reviews. The reviewers were very positive about the quality and impact of the work of the manuscript. There are some minor aspects that I would like to you to address to prepare the manuscript for publication. No new experiments are required.

In particular, I'd like you to consider reviewer 2's points:

1. Re: the title of your manuscript. Please revise the title or provide justification as to why you think it should not be.

2. Please justify why you used the time point of 24hrs of IAA treatment given the fast knockdown and consider including this justification into the manuscript.

3. You are not required to perform additional IFAs looking at the location of the suture proteins, ISC and TSCs, unless you would like to.

4. No super resolution or additional experimentation to dissect out localisation of IMC43 truncation mutants are required.

Please also address point-by-point all other queries and modify the text of the manuscript accordingly.

If all the following items are addressed, I hope to be able to make a final decision without sending the manuscript out for a second round of review.

Reviewer Comments (if any, and for reference):

Reviewer's Responses to Questions

**Part I - Summary**

Reviewer #1: The authors here present new work uncovering the functions of the alveolin proteins of Toxoplasma gondii. The authors show that conditional depletion of a novel protein, IMC43, leads to a severe defect in replication, characterised by issues in cell morphology and cell cycle control. Through elegant complementation experiments, they identify a region in the C-terminus of IMC43 required for its function, and that this region mediates interactions between IMC43 and two other IMC proteins, IMC32 and IMC44. The authors go into detail identifying the roles of IMC44 and investigate IMC32, a previously studied protein which they show here relies on IMC43 for correct localisation to daughter cell buds.

This work is very well performed and presented and provides new information about the organisation of daughter cell budding in Toxoplasma endodyogeny. The manuscript is of high quality and I believe is suitable for immediate publication. I would like to congratulate the authors on really great work.

Reviewer #2: The manuscript by Pasquarelli et al. characterizes a novel T. gondii IMC protein, IMC43. IMC43 was identified by previous IMC29 BioID experiments carried out in the Bradley lab (PMID: 36622147) and shares localization and dynamics with other markers of the daughter cell scaffold, a structure that represents the earliest daughter bud assembly.

Conditional depletion of IMC43 leads to morphological and division defects, with parasites exhibiting severely impaired proliferation. Attempts to disrupt the ORF were unsuccessful indicating that IMC43 is essential. The authors dissect IMC43 function by creating nine deletion mutants and identify a 120 aa long region at the c-terminus necessary for its function. Combining data from IMC43 BioID experiments and an IMC43 Y2H screen, the author shortlist 30 genes and follow up on two, IMC32 and the so far uncharacterized IMC44. Taking advantage of the IMC43 conditional depletion system and the identified loss of function mutant, the authors show that IMC43 is needed for correct localization of IMC32 and 44 in the mid/late budding stage. IMC43’s essential function therefore seems to be maintaining daughter bud integrity during endodoyogeny.

How daughter buds form and which genes contribute to the process is still not understood in detail, neither in T. gondii nor in Apicomplexa in general and this manuscript makes an important contribution by carefully characterizing a novel IMC protein. Overall, this is a nicely carried out study with well-controlled experiments. The presented images allow adequate interpretation of the results and the combination of BioID and Y2H screening exhibits a good strategy toward identifying interacting proteins in the parasite. The specific comments are as follows:

Reviewer #3: This is a very well executed study that characterizes the Toxoplasma gondii protein IMC43. This protein is part of the inner membrane complex of budding cells and is essential for parasite division. Toxoplasma divides by a unique mechanism known as endodyogeny and the discovery of proteins essential for the process advances the field significantly. The work presented is a thorough characterization of the localization, function, and interactors of IMC43. The experiments presented are well executed, controlled for, and carefully interpreted. The data presented is convincing and provides great insight into the scaffold that allows for parasite division. There are no major concerns about the manuscript, only a few minor details that could use clarification.

**Part II – Major Issues: Key Experiments Required for Acceptance**

Reviewer #1: No new experiments are required

Reviewer #2: 1)The authors should consider revising the title of the manuscript, as neither IMC43 nor IMC32 is functionally involved in daughter bud initiation (buds still form when proteins are conditionally depleted, e.g. Fig 1A/C, Fig S8 B/C). Although both proteins localize to the DCS, naming the complex “essential daughter bud initiation complex” may be slightly overreaching.

2) The authors use a fast acting auxin-inducible degron system and show that IMC43 is depleted after 4 hrs of IAA treatment (Fig S3B). However, the phenotypic analyses are done mostly after 24 hrs of IAA treatment, allowing parasites to undergo several rounds of division. Did the authors detect any aberrations in 4 hr IAA treated parasites that would indicate primary defects of IMC43 depletion?

3) Upon IMC43AID depletion IMC32 looses its striped pattern on daughter buds (Fig 7D/E). Have the author tested the impact of IMC43 depletion on other proteins (e.g. suture proteins ISCs and TSCs [PMID 27696623]) that have been reported to localize to similar striped patterns in daughter/mature parasites ?

Reviewer #3: None noted

**Part III – Minor Issues: Editorial and Data Presentation Modifications**

Reviewer #1: No minor points

Reviewer #2: - The authors use state of the art microscopy that clearly allows result interpretation. However, given the small size of certain parasite structures that are difficult to resolve with conventional light microscopy, have the authors considered applying super-resolution microscopy (e.g. 3D-SIM)?

This will most likely facilitate better resolution of the earliest IMC43 appearances (Fig 1G/J), where the signal seems to concentrate in distinct foci before decorating the forming daughter bud. Super-resolution imaging would further allow mapping the spatial relations of DCS constituents during bud initiation (specifically the IMC32/IMC43 relation).

- The IMC43 ∆1441-1561 deletion mutant still localizes to daughter buds although it fails to rescue the phenotype caused by IMC43AID depletion. Have the authors considered tagging the ∆1441-1561 deletion mutant with TurboID and create a BioID experiment after IMC43AID depletion? This would likely expand the list of candidates that rely on IMC43 for association with daughter buds when compared to IMC43wt BioID.

- In the IMC43AID-depleted background, does IMC44 already localize to the basal complex at bud initiation and the early budding stages, or is basal complex association timing preserved in absence of IMC43? Images provided in Fig 6E seem to correspond to the mid budding stage and do not allow this assessment.

Edits:

-line 52: change Apicomplexans to Apicomplexa

-lines 65-67: Apicomplexa use external or internal budding strategies, which should be either reflected in the sentence or “internal” be deleted.

-line 79: see above comment, I suggest deleting “internal” from this sentence and move it to the next one (lines 80-82)

-line 452: “HXGPRT” is used as abbreviation for hypoxanthine-xanthine-guanine phosphoribosyltransferase while lines 472-474 use “HPT”, this should be made uniform.

-Fig 1G: Three of the four mature parasites seem to form only a single daughter, is that correct?

-line 653; Change “Asterisk indicates daughter apical caps.” to “Asterisk indicates daughter buds”, as the asterisk does not seem to clearly point to the apical cap of the daughter.

-line 685: Please fix cross-reference for plaque assays, legends says “J” but plaque assays are shown as panel “H” of Figure 2.

-line 688: Same as above, please correct to panel “H”.

-line 691: Delete the second scale bar sentence.

-Fig 3: Please add IAA treatment time to figure legend and/or corresponding result section.

-Fig 3: Please indicate scale bar size in the figure or legend.

-Fig 3D: The - IAA “merge” image also shows the Hoechst signal in Blue, please add the corresponding Hoechst signal for the +IAA treated “merge” images as well.

-Fig 3E: This reviewer feels that the message of continues nuclear division would be stronger if Hoechst signal is also shown for images in Fig 2E, taking advantage of the longer 40 hr IAA treatment time.

-Fig 5D/Table S1: The table in the figure and the Supp. Table 1 show candidate abundance as spectral counts, while the methods (line 600) state that “candidates were ranked by normalized spectral abundance factor values…..”. This should be made uniform or alternatively, provide normalized spectral abundance factor values in Table S1.

-Fig 5D: Since the authors reference the PPKL preprint in the discussion (lines 427-428), please add a reference in the “Localization” column of the figure as well.

-Figure S2: A schematic showing where IMC43 gRNA binds and CRISPR/Cas9 double strand break occurs will enhance understanding of this figure. Please also add dimension of the scale bar to the figure or the legend.

-Figure S3: Please label western blots molecular weight with “kDa”

Reviewer #3: It is interesting that the IMC43 kd results in defects in the cytoskeleton of the mother parasite. Since IMC43 is exclusively in the daughter buds this is interesting and might warrant some discussion or speculation.

There might be a need for some clarity in line 216 where it says ‘the fusion protein is active’. Given the data presented I imagine this means that the TurboID part of the fusion is active (i.e it biotinylates). Did they test the fusion in the KD strain to know that it rescues the phenotype? That would be needed to be able to say that the IMC43 part of the fusion is active. This is not needed, but some clarity about what it is meant by the ‘fusion protein is active’ might be helpful.

PLOS authors have the option to publish the peer review history of their article (what does this mean?). If published, this will include your full peer review and any attached files.

Reviewer #1: No

Reviewer #2: No

Reviewer #3: No

Figure Files:

Data Requirements:

Reproducibility:

References:

---

## [Editor Report · Decision Letter 1]

23 Sep 2023

Dear Prof Bradley,

We are pleased to inform you that your manuscript 'Identification of IMC43, a novel IMC protein that collaborates with IMC32 to form an essential daughter bud assembly complex in Toxoplasma gondii' has been provisionally accepted for publication in PLOS Pathogens.

Best regards,

Christopher J. Tonkin

Guest Editor

PLOS Pathogens

Dominique Soldati-Favre

Section Editor

PLOS Pathogens

Kasturi Haldar

Editor-in-Chief

PLOS Pathogens

orcid.org/0000-0001-5065-158X

Michael Malim

Editor-in-Chief

PLOS Pathogens

orcid.org/0000-0002-7699-2064
---

## [Editor Report · Acceptance letter]

28 Sep 2023

Dear Prof Bradley,

We are delighted to inform you that your manuscript, "Identification of IMC43, a novel IMC protein that collaborates with IMC32 to form an essential daughter bud assembly complex in Toxoplasma gondii," has been formally accepted for publication in PLOS Pathogens.

Best regards,

Kasturi Haldar

Editor-in-Chief

PLOS Pathogens

orcid.org/0000-0001-5065-158X

Michael Malim

Editor-in-Chief

PLOS Pathogens

orcid.org/0000-0002-7699-2064